



# Enabling the use of unstructured meshes for the Large Eddy Simulation of stable atmospheric boundary layers

Ulysse Vigny [1,2], Léa Voivenel [3], Mostafa Safdari Shadloo [2,4], Pierre Bénard [2], and Stéphanie Zeoli [1]

[1]Fluid Machine Unit, University of Mons (UMONS), Mons, 7000, Belgium
[2]INSA Rouen Normandie, Univ Rouen Normandie, CNRS, Normandie Univ, CORIA UMR 6614, F-76000 Rouen, France
[3]CNRS, INSA Rouen Normandie, Univ Rouen Normandie, Normandie Univ, CORIA UMR 6614, F-76000 Rouen, France
[4]Institut Universitaire de France, Rue Descartes, F-75231 Paris, France

**Correspondence:** Ulysse Vigny (ulysse.vigny@umons.ac.be)

**Abstract.**

Modelling wind flows over complex terrain under varying atmospheric stability conditions is essential for improving our understanding of atmospheric boundary layer physics and its impact on wind energy systems. However, such simulations remain challenging due to the limitations of structured grids in representing complex geometries and the inherent difficulty of modelling the stable boundary layer, characterized by small-scale turbulent structures. These challenges necessitate the use of high-fidelity simulations with unstructured meshes, which offer greater geometric flexibility. Nevertheless, unstructured grids are rarely used in atmospheric simulations. This study establishes a baseline framework for the use of unstructured meshes in atmospheric boundary layer simulations, with particular relevance to complex terrain. The proposed solver is validated against two well-established benchmarks under neutral and stable stratification. For the neutral case, the Andrén benchmark, a $1.28 \times 1.28 \times 1.5\,\mathrm{km}^3$ periodic domain where the flow is driven by a large-scale pressure gradient, is considered. Results from structured and unstructured grids are in good agreement, with minor differences observed near the surface. Unstructured grids exhibit slightly higher friction velocities due to wall-proximal grid quality, but remain within the expected variability of existing studies. The solver is then applied to the GABLS1 stable boundary layer case, a $400 \times 400 \times 400\,\mathrm{m}^3$ domain with surface cooling. Both grid types capture the evolution of the SBL, with unstructured grids yielding higher surface heat fluxes – up to $14\%$ – resulting in a thicker boundary layer and noticeable differences in mean profiles and fluxes. A mesh refinement study confirms that a horizontal resolution of $\Delta x = 6.25\,\mathrm{m}$ is sufficient for accurate SBL representation with both mesh types. Overall, the results demonstrate that unstructured meshes are a viable and robust tool for atmospheric boundary layer modelling, capable of matching the accuracy of structured grids while offering the flexibility required for complex terrain. The minor discrepancies observed remain within the variability expected from model formulation choices. This work thus provides a foundational reference for future high-fidelity atmospheric simulations using unstructured grids, particularly in terrain-resolving contexts.





## 1 Introduction

The global push toward renewable energy, driven by urgent environmental and energy concerns, has made maximizing wind farm efficiency a critical objective (IRENA, 2019). In response, wind turbine dimensions have increased dramatically, with rotor diameters now exceeding hundreds of meters. As a result, modern wind turbines are influenced not only by micro-scale atmospheric phenomena (scales below $1\,\mathrm{km}$), but also by meso-scale processes, which span from 5 to several hundred kilometres and govern local weather systems. These turbines now operate at the intersection of micro- and meso-scale dynamics (Veers et al., 2019). The extension across scales introduces new physical mechanisms that significantly affect atmospheric flow behaviour. At higher altitudes, flow is shaped by geostrophic balance, where the pressure gradient force – arising from synoptic-scale weather systems – is countered by the Coriolis force due to Earth's rotation. Closer to the ground, the structure of the atmospheric boundary layer (ABL) is strongly modulated by thermal stratification. Solar heating induces vertical temperature gradients that give rise to buoyancy-driven forces, classifying the atmosphere as stable, neutral, or unstable depending on the gradient's direction (Kaimal and Finnigan, 1994). Near the surface, terrain-induced effects become dominant, impacting both horizontal and vertical velocity gradients and generating complex features such as flow separation, recirculation zones, and spatially varying surface roughness. Understanding these multi-scale processes is crucial for improving the modelling and prediction of wind turbine performance, especially in heterogeneous environments. Despite extensive efforts, our knowledge of atmospheric flows over complex terrain remains incomplete (Elgendi et al., 2023).

One well-established factor influencing turbine behaviour is vertical wind shear, which affects wake recovery, turbulence intensity, energy yield, and structural loads (Porté-Agel et al., 2020). While field campaigns and wind tunnel experiments offer valuable insights (Doubrawa et al., 2019; Moriarty et al., 2020), their practical application is constrained by cost and logistical complexity. Consequently, numerical simulations have become an increasingly prominent tool for studying atmospheric flows (Stoll et al., 2020). Large Eddy Simulation (LES) is currently the state-of-the-art technique for resolving turbulent structures in the ABL. While LES of the convective boundary layer (CBL) is well established (Fernando and Weil, 2010), accurately simulating the stable boundary layer (SBL) remains a significant challenge (Mahrt, 2014). The difficulty stems from the relatively small characteristic length scales of turbulent structures in stable conditions. While the CBL can extend up to $1\,\mathrm{km}$ and is dominated by large convective eddies, the SBL typically remains confined below $200\,\mathrm{m}$ and is primarily driven by wind shear. The resulting turbulence is weaker, with finer-scale vortices requiring higher spatial resolution – and, by extension, increased computational resources need – for accurate representation (Garratt, 1994). In addition to thermal effects, simulating wind flows over complex terrain introduces further challenges, particularly in mesh generation. Structured grids, which are commonly used in atmospheric simulations, often struggle to conform to intricate topographies. Unstructured meshes offer greater flexibility in handling geometric complexity and local refinement (Bates et al., 2003; Bilskie et al., 2015). However, the development of high-fidelity solvers on unstructured grids – especially for LES – remains an area of active research, and their application to realistic atmospheric flows has been limited. To help bridge this gap, the present study investigates the use of unstructured grids for LES of the atmospheric boundary layer, with a focus on thermal stratification. Simulations are performed





using both structured and unstructured meshes, enabling a direct comparison under controlled conditions. To the best of the authors' knowledge, this work represents the first LES of a stable boundary layer conducted on an unstructured mesh.

The paper is organized as follows: Section 2 outlines the numerical methodology. Section 3 presents neutral boundary layer simulations using both grid types with matching resolution. Section 4 extends the analysis to the stable boundary layer and includes a resolution sensitivity study. Final remarks and conclusions are provided in Section 5.

# 2   Methodology

## 2.1   Numerical framework

Large Eddy Simulations are performed using the incompressible, constant-density flow solver of the YALES2 platform (Moureau et al., 2011), a high-performance, finite-volume code capable of handling both structured and unstructured meshes on massively parallel architectures. The spatial discretization relies on a fourth-order central differencing scheme, while time integration is

performed using a fourth-order Runge-Kutta method (Kraushaar, 2011). Time integration is governed by a CFL condition: $CFL = U\frac{\Delta t}{\Delta x} < 0.9$.

The LES approach is based on the spatial filtering of the Navier–Stokes equations within the inertial range of turbulence. Using Einstein notation, and denoting the spatial filtering operator with a tilde ($\tilde{\bullet}$), the filtered incompressible Navier–Stokes equations are expressed as:

$$\frac{\partial \tilde{u}_j}{\partial t} + \frac{\partial \tilde{u}_i \tilde{u}_j}{\partial x_i} = \nu \frac{\partial^2 \tilde{u}j}{\partial x_i \partial x_i} + \frac{1}{\rho_0}\frac{\partial \tau ij^R}{\partial x_i} - \frac{1}{\rho_0}\frac{\partial \tilde{P}}{\partial x_j} + \frac{\tilde{\rho}g_j}{\rho_0} - 2\Omega(G_j - \tilde{u}_j) \quad \text{and} \quad \frac{\partial \tilde{u}_i}{\partial x_i} = 0 \tag{1}$$

Here, $u$ is the velocity vector, $\nu$ the kinematic viscosity, $\rho_0$ the reference air density, $\tau ij^R$ the subgrid-scale (SGS) stress tensor, and $P$ the pressure. The last two terms on the right-hand side represent buoyancy and Coriolis effects under the Boussinesq approximation (Gray and Giorgini, 1976), where $\rho$ is the density, $g$ the gravitational acceleration, $\Omega$ the Earth's angular velocity, and $G$ the geostrophic wind vector.

Subgrid-scale turbulence is modelled using the dynamic Smagorinsky model (Germano et al., 1991; Lilly, 1992), which adjusts the Smagorinsky constant based on local flow characteristics. Although more advanced models exist for capturing anisotropic turbulence (Gadde et al., 2021), the dynamic Smagorinsky model remains widely used due to its simplicity and robustness (Pope, 2001). Notably, it has demonstrated improved performance in reproducing stable boundary layer dynamics, including greater boundary layer depth compared to the original Smagorinsky formulation (Beare et al., 2006).

## 2.2   Wall model

Due to mesh resolution limitations near the surface, the near-wall region is not fully resolved. Instead, a wall model based on the Monin–Obukhov Similarity Theory (MOST) (Monin and Obukhov, 1954; Landau and Lifshitz, 1959) is employed to compute surface momentum and heat fluxes. MOST provides a unified framework capable of capturing all three stability





regimes –neutral, stable, and unstable – via correction functions in the logarithmic velocity and temperature profiles (Kaimal and Finnigan, 1994).

The velocity and temperature profiles are given by:

$$\frac{u(z)}{u_*} = \frac{1}{\kappa}\left[\ln\left(\frac{z}{z_0}\right) - \psi_m\left(\frac{z-z_0}{L}\right)\right], \tag{2}$$

$$\frac{\theta(z)-\theta_w}{\theta_*} = \frac{1}{\kappa}\left[\ln\left(\frac{z}{z_0}\right) - \psi_h\left(\frac{z-z_0}{L}\right)\right], \tag{3}$$

where $u_* = \sqrt{\tau_w/\rho}$ is the friction velocity with $\tau_w$ the local shear stress at the wall. $\theta_* = -q_w/u_*$ is the friction temperature with $q_w$ the kinematic surface heat flux and $\theta_w$ the wall temperature. $\kappa$ is the Von Kármán constant and $z_0$ the roughness length. The Obukhov length $L$ characterizes the height at which buoyancy effects begin to dominate over shear and is computed as: $L = -\frac{u_*^3 \theta_0}{\kappa g q_w}$ where $\theta_0 = 263.5\,\mathrm{K}$ is the reference potential temperature.

The correction functions $\psi_m$ and $\psi_h$ are set to zero for neutral cases, leading to a classical logarithmic velocity profile. For non-neutral configurations, they can be expressed as:

$$\psi_{m/h}(\xi) = \int_{z_0/L}^{\xi} \frac{1 - \phi_{m/h}(\xi)}{\xi}\,\mathrm{d}z, \tag{4}$$

where $\xi = z/L$ and $\phi_m$ and $\phi_h$ are termed stability functions. The latter are empirically determined depending on the stability condition. For stable cases they are expressed as:

$$\phi_m = 1 + \beta_m \xi,$$
$$\phi_h = 1 + \beta_h \xi, \tag{5}$$

Various parametrizations were introduced over the years (Businger, 1971; Högström, 1988). In this work, we use the one prescribed in the GABLS1 setup: $\beta_m = 4.8$ and $\beta_h = 7.8$.

Following the approach of Basu et al. (2008), the wall temperature is prescribed as a boundary condition rather than the surface heat flux. This introduces a coupled system with two unknowns: the friction velocity $u_*$ and the heat flux $q_w$. They are solved using a double Newton–Raphson algorithm (Ypma, 1995), selected for its quadratic convergence properties.

As the wall model is derived from filtered equations, inputs such as velocity and temperature must be appropriately averaged. For structured meshes, horizontal plane averaging at the first grid point above the surface is straightforward. In unstructured meshes, where such planes do not exist, spatial filtering is achieved using a Gather–Scatter operator applied between control volumes and nodal points (Larsson et al., 2016).

## 2.3   Grid generation

In the following studies, two grid types are used: structured (S) and unstructured (U). A constant cell size in all three directions is used. For structured meshes, this translates to a 3D Cartesian mesh with a constant cell size, using hexahedra elements.





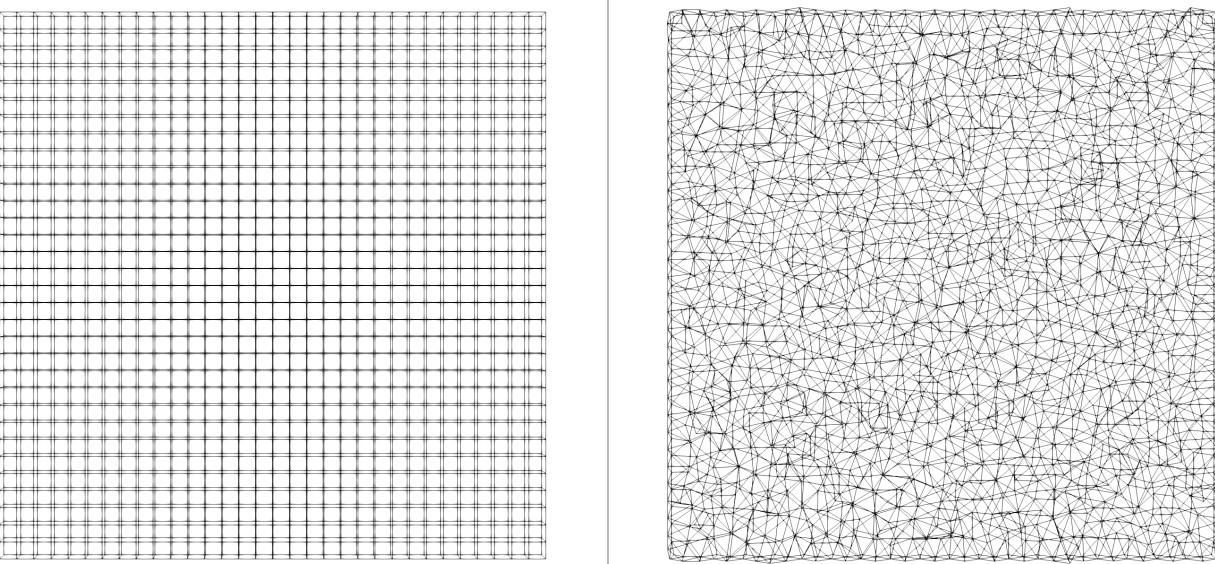

**Figure 1.** $XZ$ crinkle slice at $Y = 200\,\mathrm{m}$ for the $\Delta x = 12.5\,\mathrm{m}$ cell size mesh used in Section 4. Left: structured grid, Right: unstructured grid.

For unstructured meshes, a 3D grid using tetrahedral elements is created using the GMSH mesh generator tool (Geuzaine and Remacle, 2009). A visualisation of both meshes, based on the meshes used on Section 4, is shown in Fig. 1. Similar meshes are used on Section 3, only the domain size and the mesh size differ.

YALES2 defines control volumes at the nodes. In order to make meaningful comparisons, in the next studies the number of nodes—and therefore the degrees of freedom—between structured and unstructured meshes is similar. This translates into a significantly different number of elements depending on the element type.

Details about unstructured mesh quality can be found in Section 4.2.1. It serves as explaining some of the differences measured between the simulations obtained on the two types of mesh.

**3   Neutral boundary layer**

To establish the validity of the use of unstructured meshes for atmospheric simulations, an initial benchmark test is conducted under truly neutral stratification. For this purpose, the well-known case developed by Andren et al. (1994) is reproduced and serves as a reference.

**3.1   Case description**

The simulation domain is a rectangular box of $1280 \times 1280 \times 1500\,\mathrm{m}^3$, as illustrated by Fig. 2. Periodic boundary conditions are applied in the horizontal directions to emulate an infinite atmospheric boundary layer. A slip wall is prescribed at the domain top, while a rough bottom boundary with roughness length $z_0 = 0.1\,\mathrm{m}$ is modelled using Monin–Obukhov. The flow is driven



by geostrophic forcing, with the geostrophic wind vector set to $(G_x, G_y) = (10, 0)\,\mathrm{ms}^{-1}$, and the Coriolis parameter specified as $f = 10^{-4}\,\mathrm{s}^{-1}$. Simulations were initialized with a reference density of $\rho_0 = 1\,\mathrm{kgm}^{-3}$. The initial velocity field is defined following the original configuration from Andren et al. (1994):

$$
\begin{aligned}
u(z) &= G_x \left(1 - \exp\left(-\tfrac{z}{L_z}\right) \cos\left(\tfrac{z}{L_z}\right)\right), \\
v(z) &= G_x \exp\left(-\tfrac{z}{L_z}\right) \cos\left(\tfrac{z}{L_z}\right).
\end{aligned}
\tag{6}
$$

The simulation results are compared to four previous studies (Andren et al., 1994; Chow et al., 2005; Senocak et al., 2007; Feng et al., 2021).

The simulation is run for $30$ dimensionless time period $tf$ as in Chow et al. (2005), equivalent to $84$ physical hours. Statistical quantities are averaged over the final six dimensionless time periods, equivalent to $28\,\mathrm{h}$. It approximately corresponds to the inertial oscillation period $2\pi/f$. Senocak et al. (2007) showed that it was enough for the statistics to remain fairly stationary during the averaging period. The grid resolution is set to $\Delta = 16\,\mathrm{m}$, consistent with Feng et al. (2021) and comparable to Senocak et al. (2007). Chow et al. (2005) employed a vertically refined mesh, while Andren et al. (1994) used slightly coarser grids.

## 3.2 Results

Figure 2(b) shows the time- and horizontally-averaged streamwise velocity profile, computed over the final $28$ hours of simulation to ensure statistical stationarity. The velocity profile exhibits the expected logarithmic shape characteristic of a neutrally stratified boundary layer. This profile reflects the balance between surface friction and the geostrophic pressure gradient. Results obtained using both structured and unstructured meshes demonstrate excellent agreement with reference studies, indicating that the large-scale flow dynamics are well captured regardless of mesh type. Differences between mesh types are negligible in this case.

Quantitative comparison of the surface friction velocity, $u_*$, is provided in Table 1. For the structured mesh simulation, $u_* = 0.409\,\mathrm{ms}^{-1}$, while the unstructured mesh yields a slightly higher value of $u_* = 0.438\,\mathrm{ms}^{-1}$. These results fall within the range reported in the four prior studies. The slightly elevated $u_*$ observed in the unstructured simulation can be attributed to increased numerical diffusion near the wall, a known characteristic of unstructured meshes – especially in double-periodic domains where mesh regularity is harder to maintain – but remains into the literature values range. Lower mesh quality in the near-wall region can amplify momentum transfer, thereby increasing surface stress and the corresponding friction velocity.

To further assess flow characteristics, Fig. 3 shows vertical profiles of the streamwise velocity variance $\overline{u'^2}$ (left) and the Reynolds shear stress $\langle \overline{u'w'} \rangle$ (right). Results from both structured and unstructured mesh simulations are compared to the spread reported in the reference studies. In both cases, the profiles fall within the variability range found in the literature. However, the unstructured mesh simulation exhibits slightly higher velocity variance near the wall. This is consistent with the previously observed higher surface friction velocity, as enhanced near-surface shear naturally leads to stronger velocity fluctuations. While these differences are measurable, they remain smaller than the broader inter-study variability, which arises from differing numerical schemes, subgrid-scale models, and wall treatment methods.



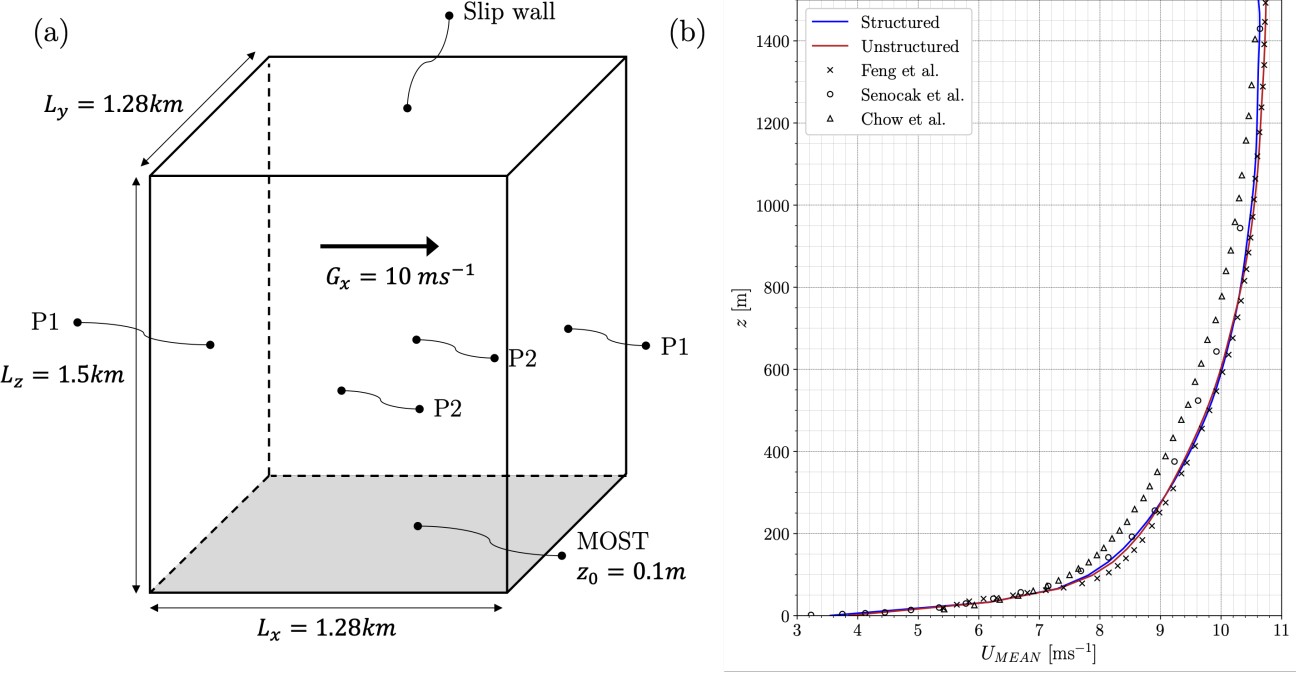

**Figure 2.** (a): Andrén setup configuration scheme. P1 and P2 for periodic boundaries in pairs. (b): Average velocity profile of the Andrén case gathered on 28 h.

**Table 1.** Frictional velocity for the four different studies. Andren/Moeng, Mason/Brown, Nieuwstadt, and Schumann/Graf refers to the four distinct LES codes used in Andren et al. (1994).

| Study | $u_* \, [ms^{-1}]$ |
|---|---|
| Feng et al. | 0.419 |
| Chow et al. | 0.44 |
| Senocak et al. | 0.42 |
| Andren/Moeng | 0.425 |
| Mason/Brown bsct | 0.448 |
| Mason/Brown nbsct | 0.402 |
| Nieuwstadt | 0.402 |
| Schumann/Graf | 0.425 |
| Current work – Structured | 0.409 |
| Current work – Unstructured | 0.438 |





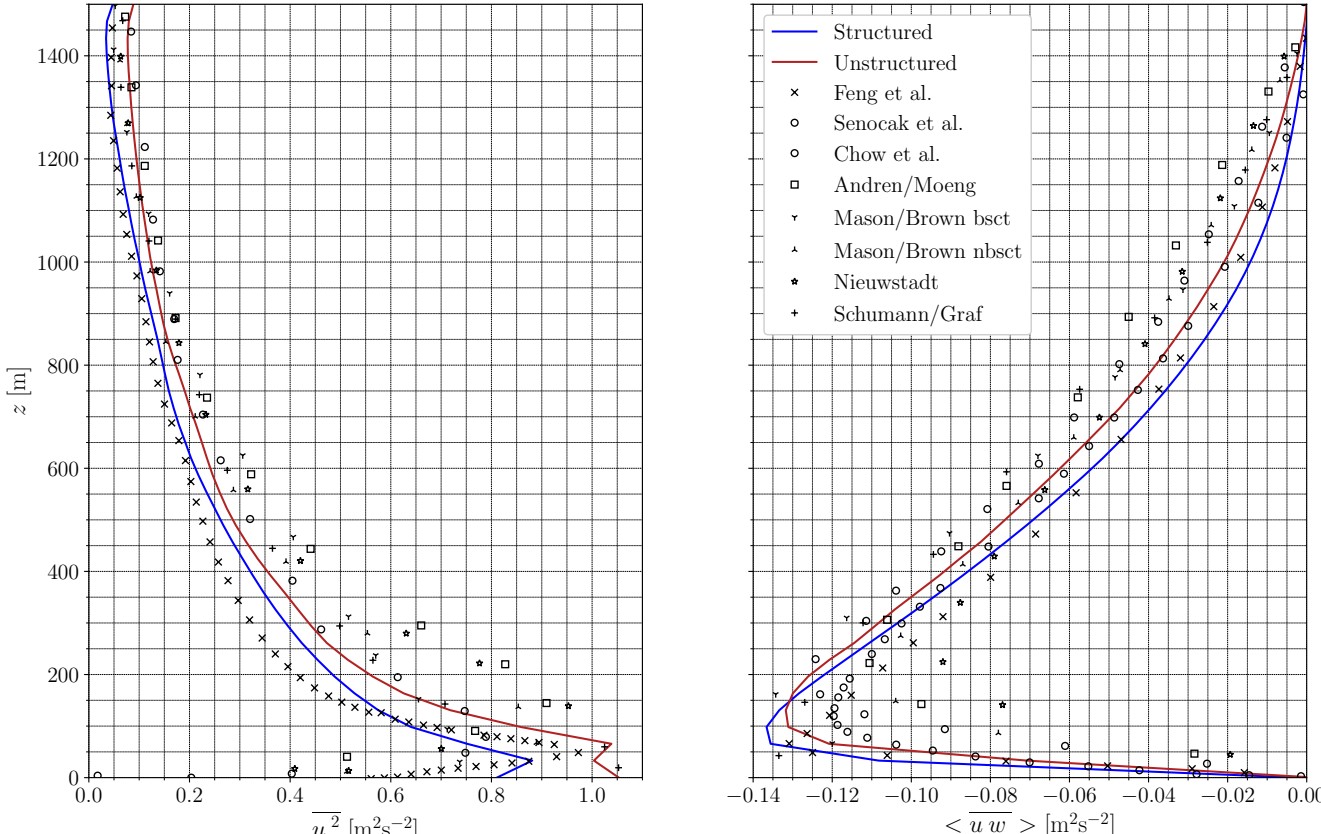

**Figure 3.** Average streamwise velocity variance and momentum flux profile of the Andrén case gathered on 28 h.

This simple neutral configuration serves as a first validation step for the proposed methodology. The results demonstrate that the methodology using unstructured meshes is capable of accurately reproducing the dynamics of a neutral atmospheric boundary layer. All key flow quantities fall within the range of previous studies, confirming the robustness of the approach. No mesh sensitivity study has been conducted for this neutral case. As turbulent structures are relatively large and easier to resolve, a fine mesh is unnecessary. However, the results still highlight the influence of the mesh, particularly near the wall. The next step focuses on the stable boundary layer, where thermal stratification introduces additional physical challenges and where mesh resolution plays a more critical role in capturing the finer turbulent structures.

## 4 Stable boundary layer

To improve understanding of the stable atmospheric boundary layer and its representation in LES, the Global Energy and Water Cycle Experiment (GEWEX) launched the GEWEX Atmospheric Boundary Layer Study (GABLS)(Holtslag et al., 2012). The GABLS initiative has focused on land-based SBLs and the accurate simulation of their diurnal cycle. Over time,





three benchmark intercomparison cases have been defined, progressively increasing in realism(Sanz Rodrigo et al., 2017). This study adopts the first intercomparison case, known as GABLS1, which targets an idealized Arctic SBL scenario (Kosović and Curry, 2000).

## 4.1 Case description

The original GABLS1 intercomparison involved 11 LES codes and demonstrated that accurate resolution of the SBL strongly depends on mesh resolution (Beare et al., 2006). Follow-up studies have employed the GABLS1 setup to investigate the influence of grid refinement (Sullivan et al., 2016; Min and Tombouldies, 2022), surface cooling rates (Sullivan et al., 2016; Kumar et al., 2010; Huang and Bou-Zeid, 2013), and subgrid-scale modelling strategies (Matheou and Chung, 2014; Ghaisas et al., 2017; Gadde et al., 2021). Additional research has used GABLS1 as a validation benchmark for Reynolds-Averaged
Navier–Stokes (RANS) models and pseudo-spectral LES methods (Sanz Rodrigo et al., 2017; Lazeroms, 2015). While previous studies have successfully reproduced key SBL characteristics, they have all employed structured meshes. To the best of the authors' knowledge, no study has yet explored the use of unstructured grids for this configuration, likely due to the relatively simple geometry of the domain. In this work, we assess the capability of unstructured meshes to replicate the GABLS1 benchmark results, and we compare them directly against structured-mesh simulations at matched spatial resolutions.

The computational domain measures $400 \times 400 \times 400 \, \mathrm{m}^3$, with periodic boundary conditions in the streamwise ($x$) and spanwise ($y$) directions. The bottom boundary is a rough wall, with a surface roughness length of $z_0 = 0.1 \, \mathrm{m}$ and a fixed surface temperature $T_w = 265 \, \mathrm{K}$. A constant cooling rate of $0.25 \, \mathrm{Kh}^{-1}$ is imposed. The top boundary is treated as a stress-free slip wall. A geostrophic wind of $G_x = 8 \, \mathrm{ms}^{-1}$ is applied in the $x$-direction. A Coriolis parameter of $f = 1.39 \times 10^{-4} \, \mathrm{s}^{-1}$ is prescribe, corresponding to a latitude of 73°N. Other physical parameters include gravity $g = 9.81 \, \mathrm{ms}^{-2}$, reference potential
temperature $\theta_0 = 263.5 \, \mathrm{K}$, air density $\rho_0 = 1.3223 \, \mathrm{kgm}^{-3}$, and the von Kármán constant $\kappa = 0.4$. Figure 4 illustrates the domain configuration and initial profiles.

The initial velocity field is set to the geostrophic wind throughout the domain. The temperature profile is initialized at $265 \, \mathrm{K}$ in the lower $100 \, \mathrm{m}$ and increases linearly by $0.01 \, \mathrm{Km}^{-1}$ above that height, reaching $268 \, \mathrm{K}$ at the top. To initiate turbulence, random perturbations of amplitude $0.1 \, \mathrm{K}$ are added to the temperature field within the lowest $50 \, \mathrm{m}$, as described in Beare et al.
(2006).

To damp gravity wave reflection near the upper boundary, a sponge layer is implemented above $300 \, \mathrm{m}$. This layer relaxes velocity and temperature fields toward their initial target profiles using the formulation:

$$SL_\phi = \gamma \sin^2 \left( \frac{z - Z_{SL}}{L_z - Z_{SL}} \frac{\pi}{2} \right) (\phi_{target} - \phi), \qquad (7)$$

where $\phi$ represents either the velocity or temperature, $\phi_{target}$ denotes the target (geostrophic or stratified) profile, and $\gamma = 1/5$
is a time relaxation parameter. The vertical range of the sponge layer extends from $z_{SL} = 300 \, \mathrm{m}$ to the domain top at $z_{top} = 400 \, \mathrm{m}$.

For stable boundary layer simulations, the CFL is modified in accordance with shallow water models (Walters et al., 2009). Due to the explicit integration of the Coriolis force, the time step is here chosen following the approach of Audusse et al.





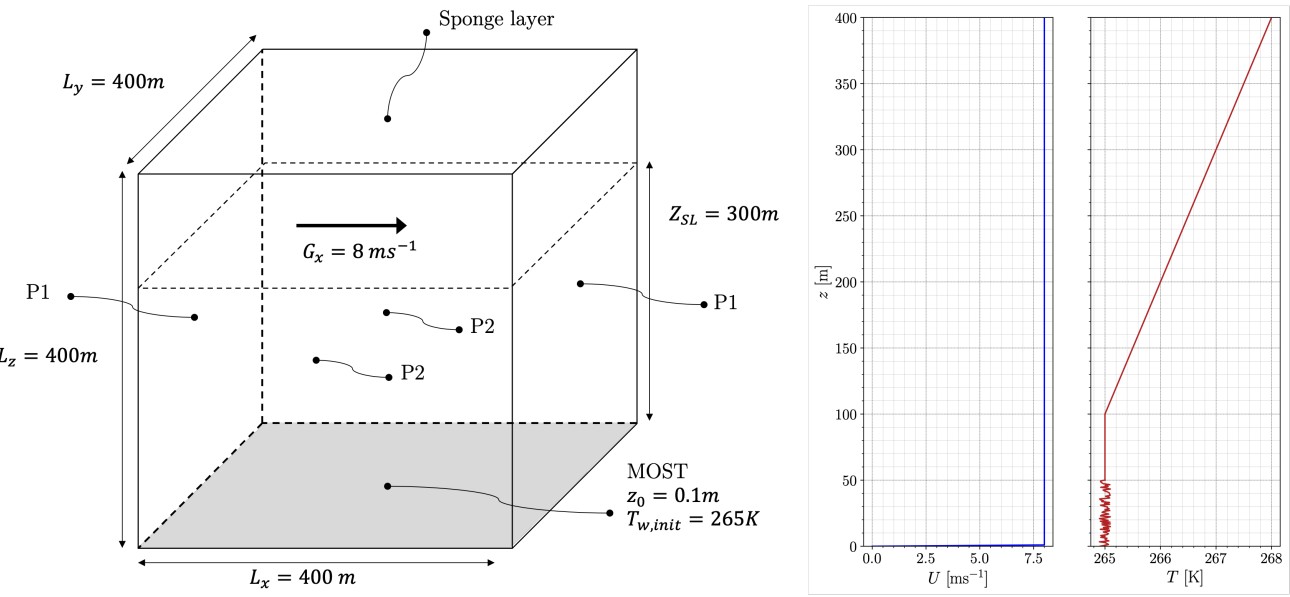

**Figure 4.** (a): GABLS1 setup configuration scheme. $P1$ and $P2$ for periodic boundaries in pairs. (b): initial velocity and temperature vertical profiles.

(2018) such that $\Delta t = \frac{CFL \times \Delta x}{\|U\| + \sqrt{gH}}$ with $H$ the vertical depth of the fluid, i.e. the stable boundary layer height. The CFL condition

remains $CFL < 0.9$. Convective velocity and boundary layer height are chosen in accordance with the a priori values of the upcoming simulations: $\|U\| = 9\,\mathrm{ms}^{-1}$ and $H = 200\,\mathrm{m}$ respectively. All time steps used in this study are summarized in Table 2.

Simulations are run for a total of $8$ hours of physical time, corresponding to a full diurnal cycle. Statistics are gathered during the final hour of simulation (between hour 7 and 8), once quasi-stationary conditions are achieved. Results are compared with the original GABLS1 intercomparison (Beare et al., 2006), as well as a wide range of follow-up studies (Kumar et al., 2010;

Huang and Bou-Zeid, 2013; Matheou and Chung, 2014; Sullivan et al., 2016; Abkar and Moin, 2017; Ghaisas et al., 2017; Gadde et al., 2021; Dai et al., 2021).

Structured and unstructured meshes at four different spatial resolutions are employed. Table 2 summarizes the grid sizes, number of elements, number of nodes, and the corresponding time step. As stated in Section 2.2, to ensure a fair comparison between mesh types, the number of nodes is comparable.

## 4.2 Results

### 4.2.1 Unstructured and Structured grid comparison

Following the GABLS1 recommendations (Beare et al., 2006), a cell size of $\Delta x = 3.125\,\mathrm{m}$ is used for both mesh types. The structured and unstructured grids are referred to as $S3$ and $U3$, respectively (see Tab. 2).





**Table 2.** Case set-up with $\Delta x$ the mesh cell size, $N_{elem}$ the number of mesh elements, $N_{node}$ the number of mesh nodes and $\Delta t$ the time step.

| Mesh name | S1 | U1 | S2 | U2 | S3 | U3 | S4 | U4 |
|---|---|---|---|---|---|---|---|---|
| $\Delta x\,[\mathrm{m}]$ | 12.5 | | 6.25 | | 3.125 | | 2.0 | |
| $N_{elem}\,[\times 10^3]$ | 32.8 | 148.2 | 262.1 | 1186 | 2097.2 | 9487.9 | 8000.0 | 35972.8 |
| $N_{node}\,[\times 10^3]$ | 45.4 | 41.8 | 366.3 | 337.8 | 2919.7 | 2659.1 | 11255.5 | 10042.7 |
| $\Delta t\,[\mathrm{s}]$ | 0.2 | | 0.1 | | 0.05 | | 0.032 | |

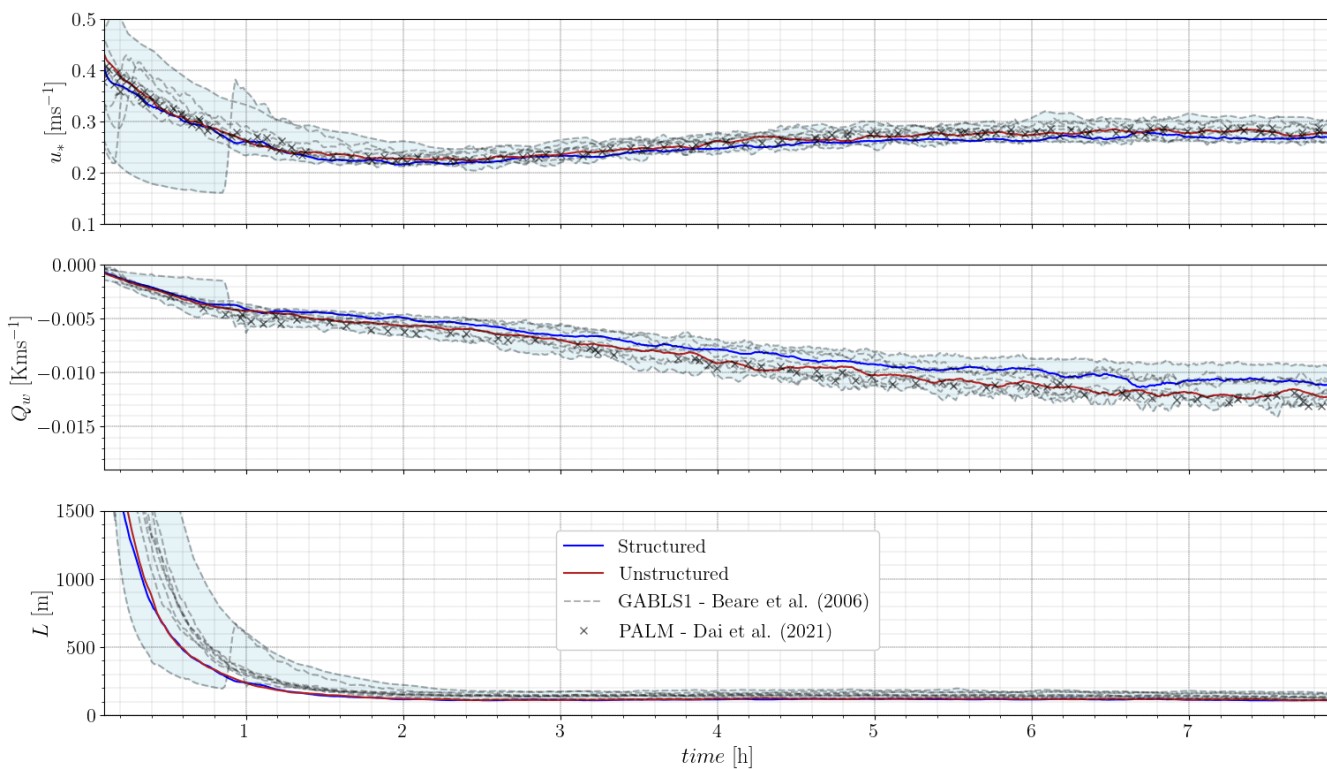

**Figure 5.** Frictional velocity, wall heat flux and Monin-Obukhov length with $S3$ and $U3$ grids, compared to original GABLS1 results dispersion (Beare et al., 2006) and PALM results (Dai et al., 2021).

Figure 5 shows the temporal evolution of key surface-layer quantities: friction velocity ($u_*$), wall heat flux($Q_w$), and Monin–Obukhov length ($L$). All time series lie within the range of the GABLS1 intercomparison dataset (Beare et al., 2006), indicating overall agreement with previous LES studies. The friction velocity is similar for both grids throughout the simulation, with $S3$ producing slightly lower values than $U3$. This observation is consistent with results in the neutral boundary layer and indicates that the numerical diffusion associated with grid type remains modest. Notably, the $U3$ simulation closely matches results from the PALM model (Dai et al., 2021), a widely adopted LES code in atmospheric modelling.





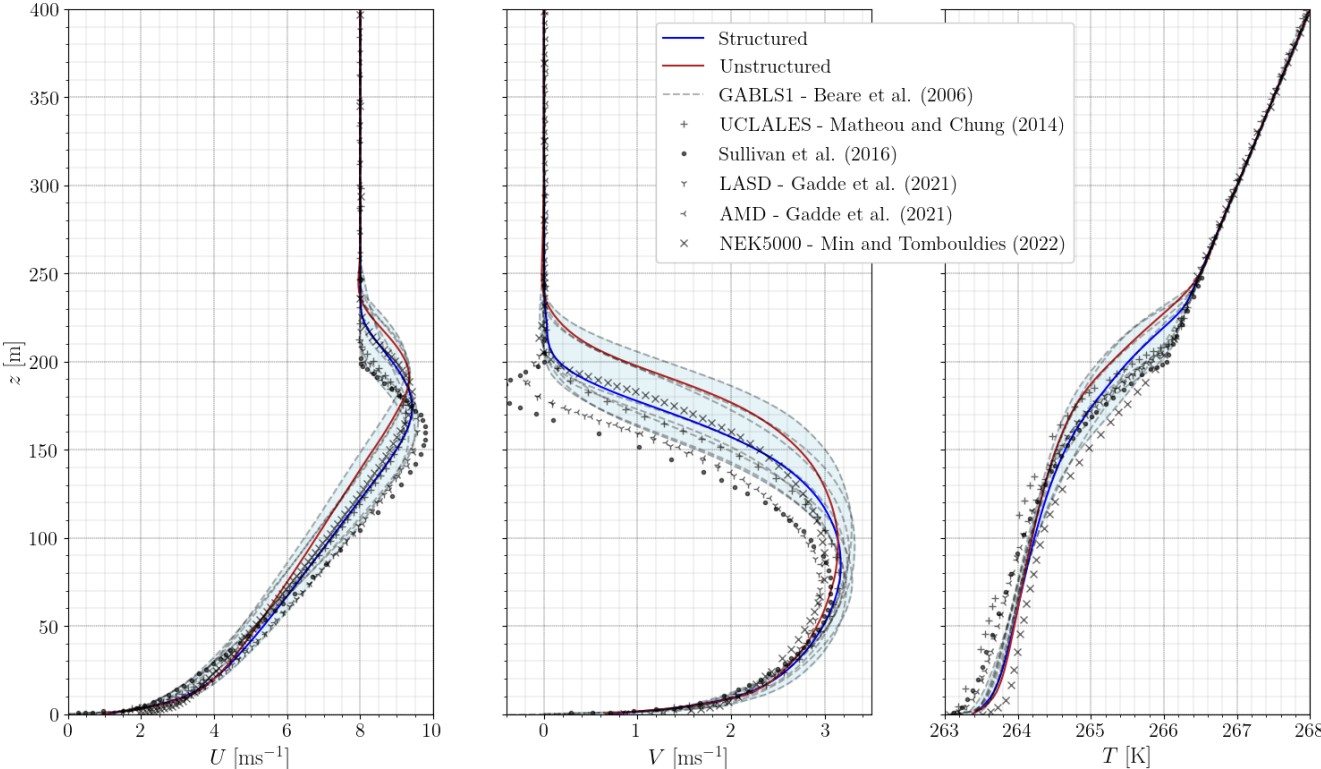

**Figure 6.** Time- and horizontally-averaged streamwise velocity, tangential velocity and temperature profiles, for meshes $S3$ and $U3$ with cell size $\Delta x = 3.125\,\mathrm{m}$. Blue shaded area stands for the original GABLS1 study results dispersion (Beare et al., 2006) and symbols for more recent studies (Matheou and Chung, 2014; Sullivan et al., 2016; Gadde et al., 2021; Min and Tombouldies, 2022).

The Monin–Obukhov length exhibits similar trends for both meshes. After an initial rapid decrease due to surface cooling, $L$ stabilizes at $O(100)\,\mathrm{m}$. Wall heat fluxes show a general decay over time in both cases. However, a gap develops after the first hour, reaching a maximum difference of approximately $14\%$ between the seventh and eighth simulation hour. This divergence likely reflects differences in near-wall numerical behaviour caused by mesh structure.

In addition to this mesh-induce difference, another source of error could be the initial temperature perturbations. Indeed, after initialization, the flow destabilizes and transitions from a laminar to a turbulent stable boundary layer. This process is sensitive to the initial temperature perturbations, which differ slightly across simulations due to the imposed random fields. This sensitivity is further discussed in Appendix 5. It could contributes to divergence among simulations. Some of the larger excursions observed in the original GABLS1 ensemble (see Fig. 5) may result from similar effects.

Time- and horizontally-averaged profiles of streamwise velocity $U$, tangential velocity $V$, and potential temperature $\theta$ are shown in Fig. 6. Averaging is performed between the 7th and 8th simulation hour, following the GABLS1 protocol. The velocity profiles exhibit a well-formed stable boundary layer, with a low-level jet located between $160 - 200\,\mathrm{m}$. The tangential





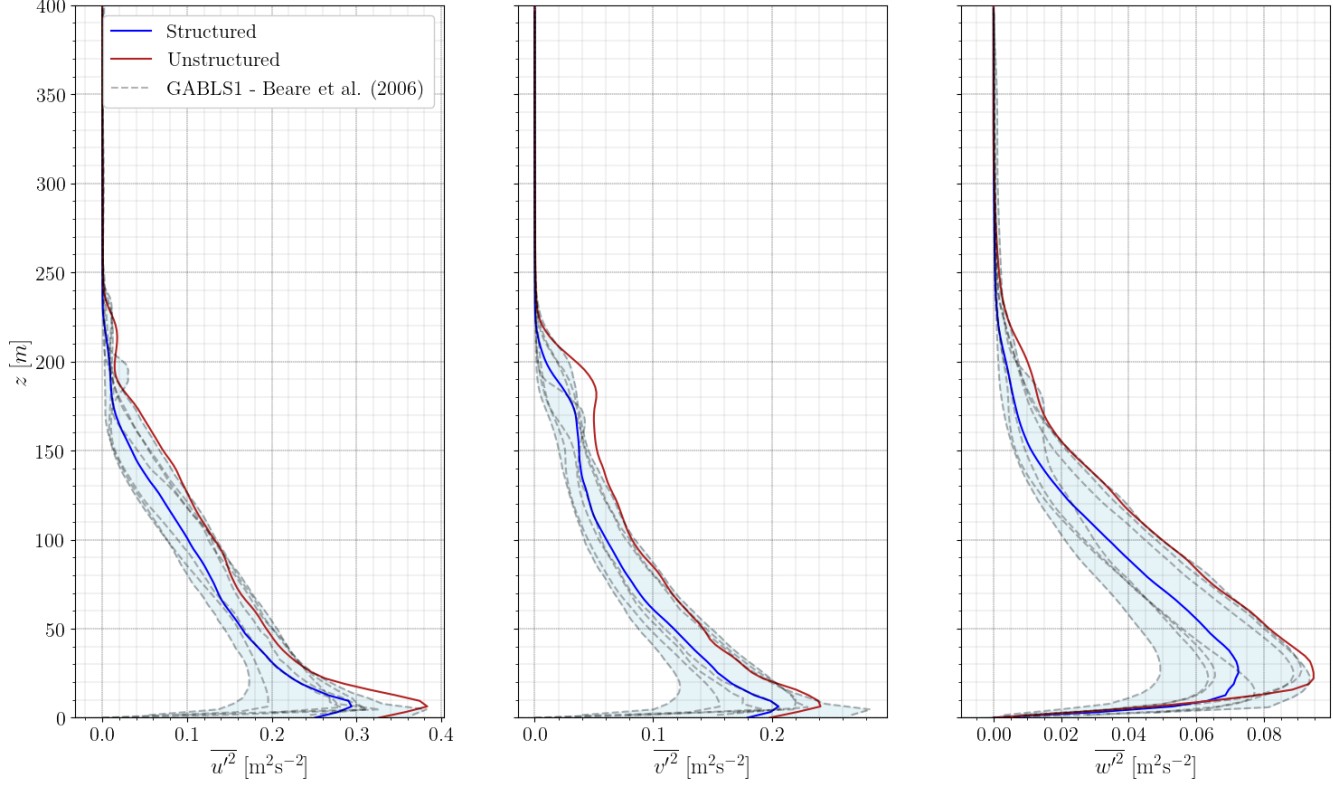

**Figure 7.** Time- and horizontally-averaged streamwise, tangential and vertical velocity variances profiles with $S3$ and $U3$ of cell size $\Delta x = 3.125\,\mathrm{m}$. Blue shading stands for the original GABLS1 study results dispersion (Beare et al., 2006).

velocity grows with decreasing height due to Coriolis acceleration, and vanishes at the surface under the action of surface drag. The temperature increases with height, with a distinct inversion layer forming between $150\,\mathrm{m}$ and $200\,\mathrm{m}$.

Differences between $S3$ and $U3$ results are visible but fall within the GABLS1 spread. The $U3$ simulation shows a $20\,\mathrm{m}$ upward shift in the velocity peak and corresponding temperature inflection point, indicating a slightly deeper or displaced SBL structure. Similar variations are found in more recent LES studies, which also broaden the envelope of acceptable solutions. For instance, the simulation by Sullivan et al. (2016) shows a negative tangential velocity near the top of the SBL that remains unexplained. Such discrepancies underscore the challenge of defining unique reference data for the SBL, but also validate the fidelity of both $S3$ and $U3$ configurations.

Figure 7 displays the time- and horizontally-averaged velocity variances for all three components. Variances are higher in the lowest part of the boundary layer – the region with turbulence production – except near the surface where they are suppressed by the wall model. In the geostrophic region, turbulent motions decrease to zero defining a zone comparable to the free atmosphere. Across all velocity components, the $U3$ mesh exhibits consistently higher variance levels than $S3$. This





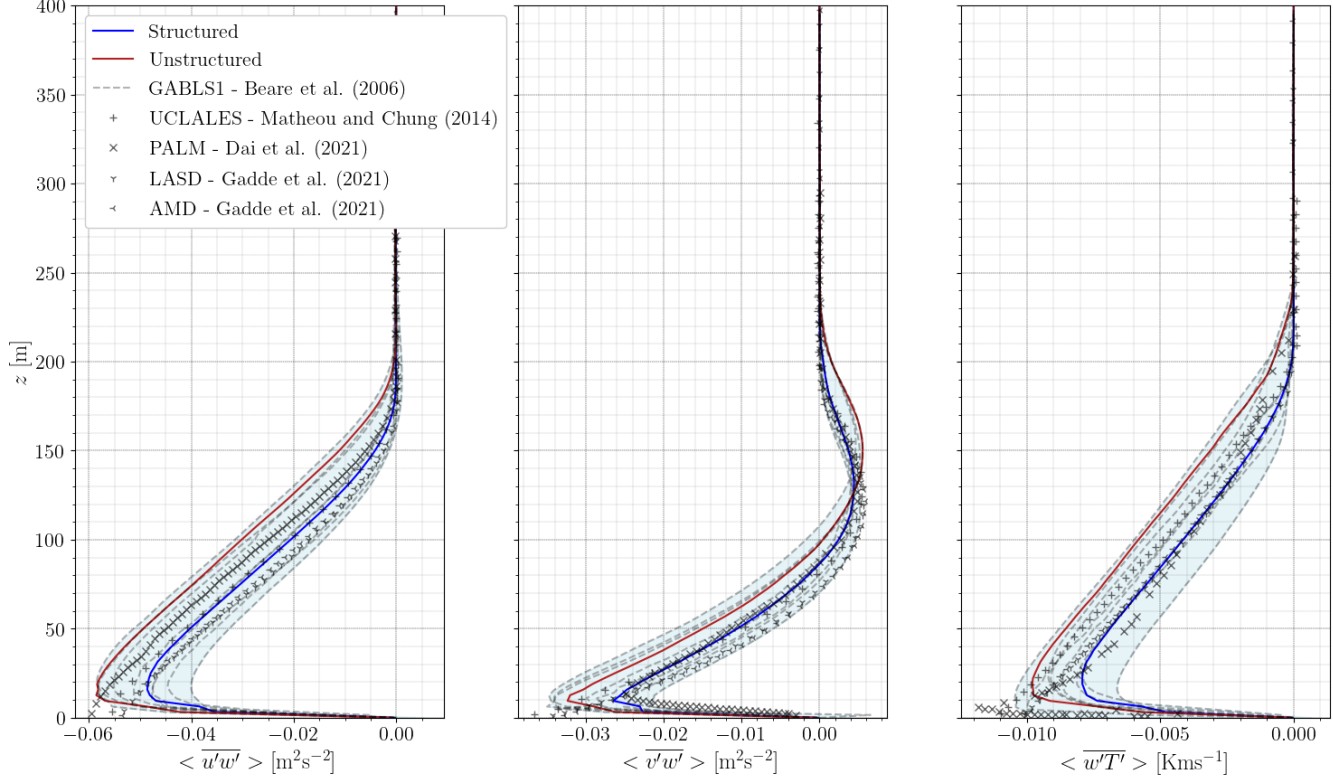

**Figure 8.** Time- and horizontally-averaged momentum and heat fluxes profiles for meshes $S3$ and $U3$ with cell size $\Delta x = 3.125\,\mathrm{m}$. Blue shading stands for the original GABLS1 study results dispersion (Beare et al., 2006) and symbols for more recent studies (Matheou and Chung, 2014; Dai et al., 2021; Gadde et al., 2021).

reflects increased resolved turbulence, which can be attributed to differences in near-wall resolution and numerical dissipation
in unstructured grids.

The time- and horizontally-averaged momentum fluxes $\langle \overline{u'w'} \rangle$, $\langle \overline{v'w'} \rangle$ and heat flux $\langle \overline{w'T'} \rangle$ are plotted in Fig. 8. All fluxes follow expected vertical trends and are consistent with past studies. As with the velocity variances, the $U3$ simulation shows a vertical offset associated with greater turbulent transport. Again this difference is likely due to stronger fluctuations resolved near the wall.

In summary, both $S3$ and $U3$ grids reproduce the SBL structure of the GABLS1 case accurately. Quantities such as friction velocity, velocity and temperature profiles, variances, and fluxes remain within the intercomparison ensemble spread. Nonetheless, systematic differences appear between the two grid types, particularly in wall heat flux, velocity variance, and flux magnitude.

These discrepancies may arise from three primary sources:





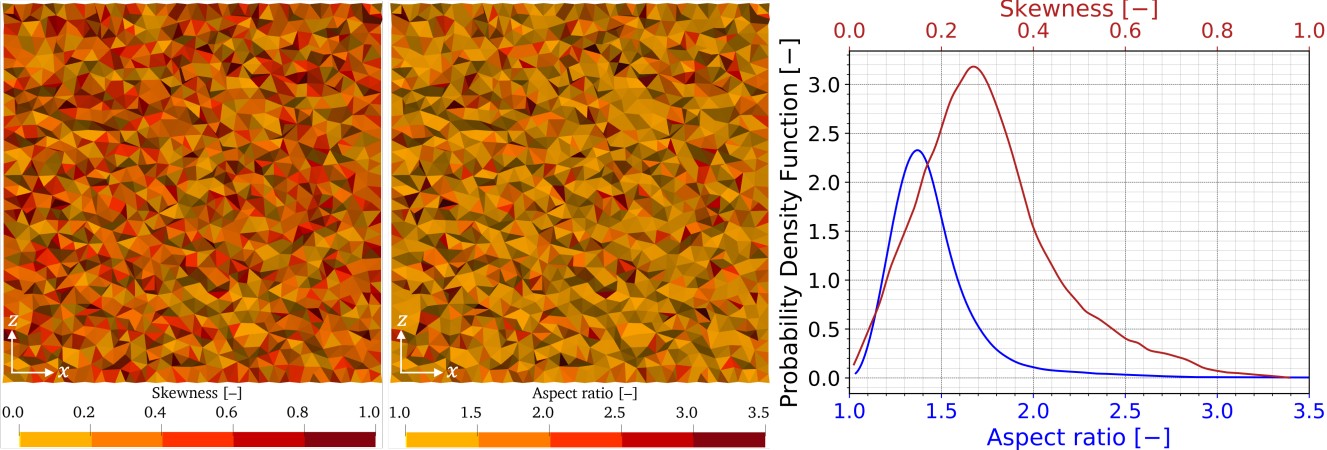

**Figure 9.** From left to right: $XZ$ skewness plane at $Y = 200\,\mathrm{m}$, $XZ$ aspect ratio planes at $Y = 200\,\mathrm{m}$, probability density function (PDF) of the skewness and the aspect ratio distribution for $U3$ mesh.

- – Grid quality: Although $S3$ and $U3$ share the same nominal resolution, the unstructured mesh introduces local geometric irregularities. Fig. 9 shows the skewness and aspect ratio distribution through the probability density function (PDF) and illustrate the spatial distribution using a 2D plane. Most $U3$ cells have acceptable quality (global mean skewness of $0.3$), but up to $0.5\%$ of cells exceed a skewness of $0.8$ and can even reach locally values of $0.96$. The aspect ratio follow a similar trend, with most element with an aspect ratio close to one, but up to $0.5\%$ of cells exceed an aspect ratio of $3$.

- – Numerical scheme accuracy: YALES2 employs a fourth-order finite-volume scheme for structured meshes. On unstructured grids, however, discretization accuracy may degrade to third order locally due to mesh geometric non-uniformity.

- – Flux estimation near walls: Wall fluxes are particularly sensitive to mesh quality and orientation. Irregular cell faces in $U3$ can increase inaccuracy in heat and momentum transfer calculations.

Despite these sources of variability, both structured and unstructured simulations yield physically consistent and accurate representations of the stable boundary layer. Differences between grid types are small and comparable to the inter-model variability reported in the literature. This supports the use of unstructured meshes for atmospheric LES in wind energy contexts, provided grid quality and wall treatment are carefully considered.

### 4.2.2 Sensitivity to resolution

Following the comparison of structured and unstructured meshes at the recommended resolution, we now investigate the influence of grid resolution on the simulation results. A sensitivity study is conducted with mesh sizes ranging from $\Delta x = 12.5\,\mathrm{m}$ to $\Delta x = 2\,\mathrm{m}$. Simulations are labelled $S1$ to $S4$ and $U1$ to $U4$ for structured and unstructured grids, respectively, with increasing index indicating higher resolution as listed in Table 2.





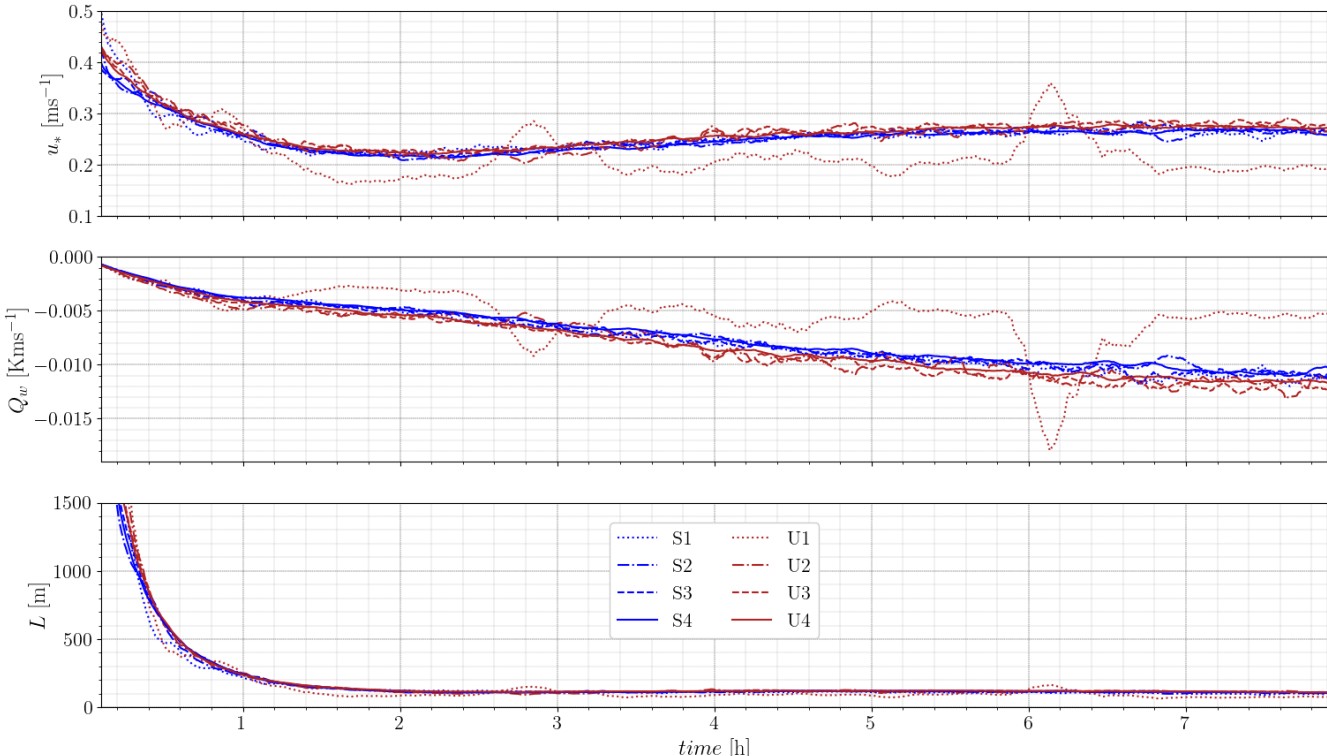

**Figure 10.** Friction velocity, wall heat flux, and Monin–Obukhov length time series.

Time series of surface-integrated quantities are shown in Fig.10. All simulations exhibit similar trends, with the exception of $U1$, which produces unreliable results. The friction velocity time series for structured grids is slightly lower and exhibits more noise. More significant differences are observed in the wall heat flux, which is consistently higher for unstructured grid simulations. These differences influence the resulting temperature, velocity, and flux profiles discussed in Section 4.2.1. However, as the mesh is refined, results from both grid types converge, though not identically. Using finer resolutions reduces numerical diffusion and enables improved gradient capture. This supports the hypothesis that the discrepancy between grid types arises primarily from resolution-driven numerical effects. For the Monin–Obukhov length – a global measure of stability – all simulations converge closely.

Figures 11 and 12 provide a qualitative illustration of flow structures across resolutions. Both figures show instantaneous velocity and temperature fields in planes perpendicular to the $Y$ axis. As expected, mesh refinement reveals progressively finer-scale turbulent structures. While the resolution level has a visible impact, differences between structured and unstructured grids are not apparent at first glance.

Horizontally and temporally averaged profiles between the 7th and 8th simulation hour are showed on Fig. 13. Except for $U1$, all cases follow similar trends. The temperature inflection point for unstructured grids consistently lies above that of the structured grids, confirming observations from Section 4.2.1. Velocity components tend to return to geostrophic values aloft.

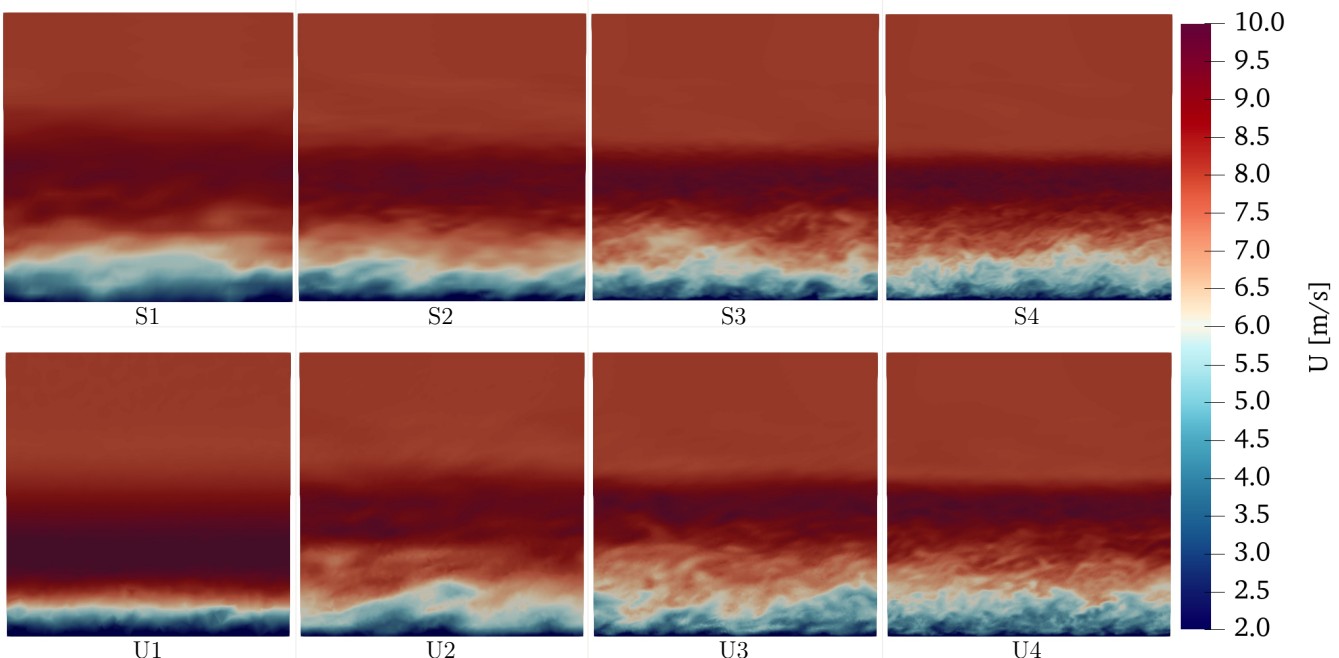

**Figure 11.** $XZ$ velocity planes at $Y = 200\,\mathrm{m}$. Top: structured cases, bottom: unstructured cases. From left to right mesh resolution increases.

Figure 14 displays momentum and heat flux profiles. For all but the coarsest mesh, unstructured grids yield stronger fluxes throughout the boundary layer. At coarsest resolution, fluxes are weak or vanish, indicating poor boundary layer representation.
Finer meshes lead to stronger turbulent fluctuations, as expected from reduced numerical dissipation. These stronger fluxes correlate with the enhanced wall heat flux observed in unstructured simulations. Overall, except for the $U1$ and $S1$ cases, all other simulations produce satisfactory results. The coarse resolution in these two cases leads to unrealistic boundary layer development and are therefore excluded from further analysis.

To further assess simulation quality, the boundary layer height is computed following the approach proposed in Kosović
and Curry (2000), based on the vertical distribution of turbulent stress. The SBL height is defined as the altitude where the tangential turbulent stress drops to $\alpha = 5\%$ of its surface value. A linear extrapolation is applied:

$$h = \frac{z\big|_{\langle \overline{u'w'} \rangle = \alpha u_*^2}}{1 - \alpha}. \tag{8}$$

Table 3 reports SBL heights from the present work and prior studies. Across all cases, boundary layer height decreases with increasing resolution, converging around $160 - 175\,\mathrm{m}$. Unstructured grid simulations yield SBL heights about $10\%$ higher,
consistent with their stronger surface heat flux and momentum fluxes (Fig. 14).

The original GABLS1 study used $\Delta x = 1\,\mathrm{m}$ as a reference, yielding an SBL height of $157\,\mathrm{m}$. Simulations within $20\%$ of this value are considered acceptable (Beare et al., 2006). By this standard, all simulations with $\Delta x \le 6.25\,\mathrm{m}$ are accurate, with deviations between $3.8\%$ and $1.9\%$ for structured grids and $18\%$ to $14\%$ for unstructured grids.





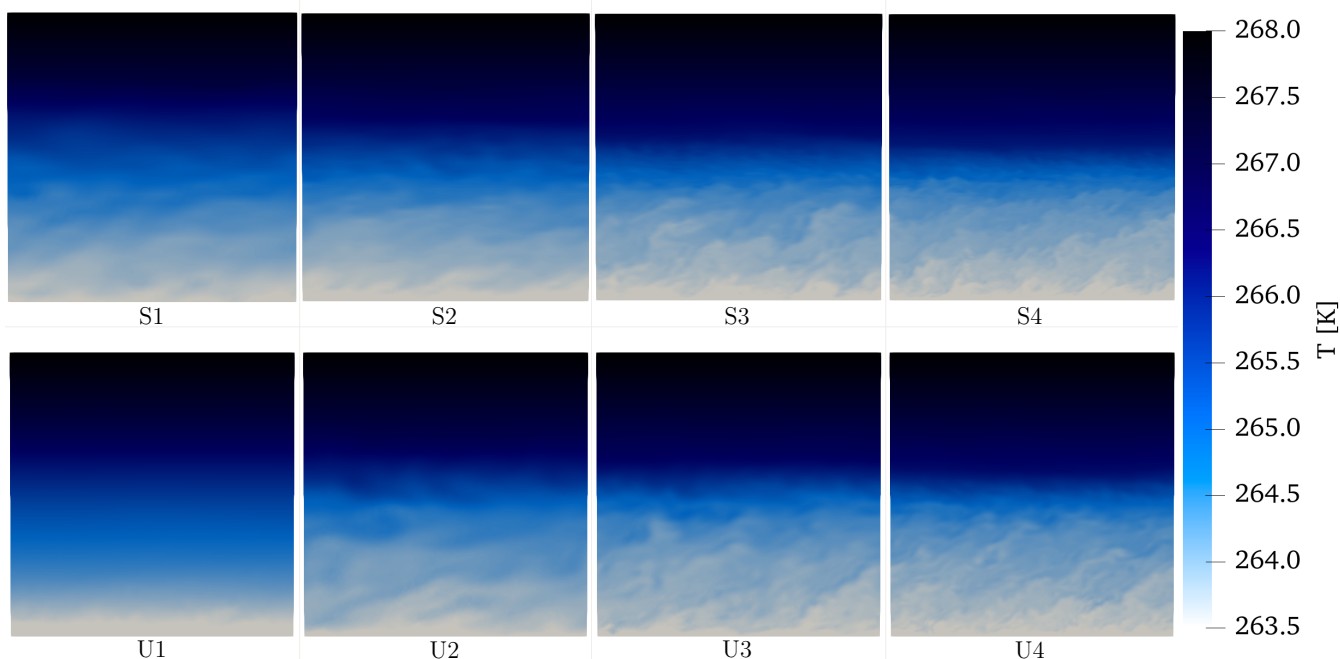

**Figure 12.** $XZ$ temperature planes at $Y = 200\,\mathrm{m}$. Top: structured cases, bottom: unstructured cases. From left to right mesh resolution increases.

**Table 3.** Boundary layer heights in various studies, depending on the grid resolution.

| $\Delta x\,[\mathrm{m}]$ | 12.5 | 6.25 | 3.125 | 2 |
|---|---|---|---|---|
| GABLS1 (Beare et al., 2006) | 215 | 188 | 182 | 174 |
| Cuxart et al. (Cuxart et al., 2006) - LES | - | - | 177 | - |
| Stoll and Porté-Agel (Stoll and Porté-Agel, 2008) | - | - | 173 | - |
| Huang and Bou-Zeid (Huang and Bou-Zeid, 2013) | - | - | - | 158 |
| Abkar and Moin (Abkar and Moin, 2017) | 168 | 165 | 169 | - |
| Gadde et al. (Gadde et al., 2021) | - | - | - | $166 - 176$ |
| Min et al. (Min and Tombouldies, 2022) | - | - | 160 | - |
| Current work - Structured | 149 | 163 | 162 | 161 |
| Current work - Unstructured | 149 | 180 | 186 | 179 |

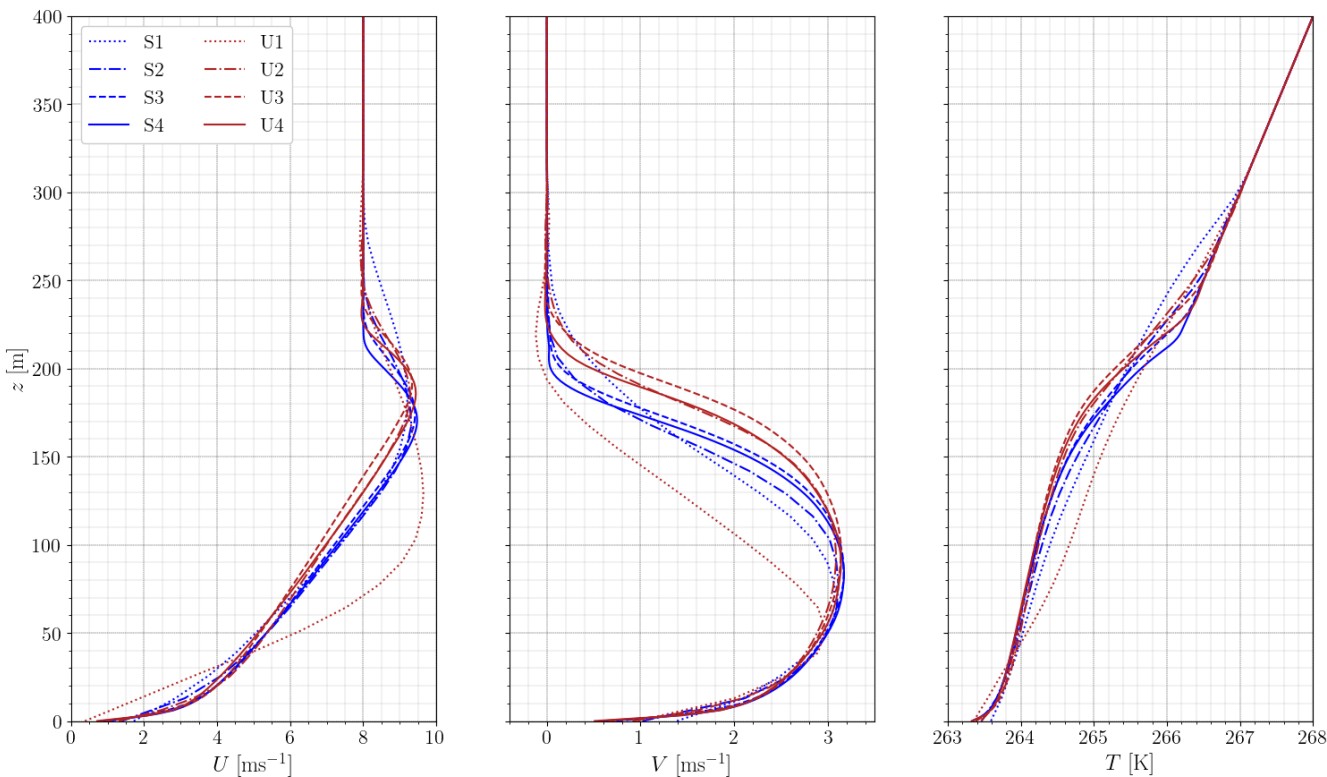

**Figure 13.** Time- and horizontally-averaged streamwise velocity, tangential velocity and temperature profiles.

**Table 4.** Relative $L2$ norm error in % of the horizontal average velocity and temperature profiles compared to the reference profiles from Beare et al. (2006).

| Mesh type | Quantity | $\Delta x \, [\mathrm{m}]$ | | | |
|---|---|---|---|---|---|
| | | 12.5 | 6.25 | 3.125 | 2 |
| Unstructured | $\langle U \rangle$ | 9.2 | 4.8 | 5.9 | 4.6 |
| | $\langle T \rangle$ | 0.09 | 0.08 | 0.09 | 0.07 |
| Structured | $\langle U \rangle$ | 5.3 | 2.9 | 2.7 | 1.9 |
| | $\langle T \rangle$ | 0.09 | 0.06 | 0.05 | 0.03 |





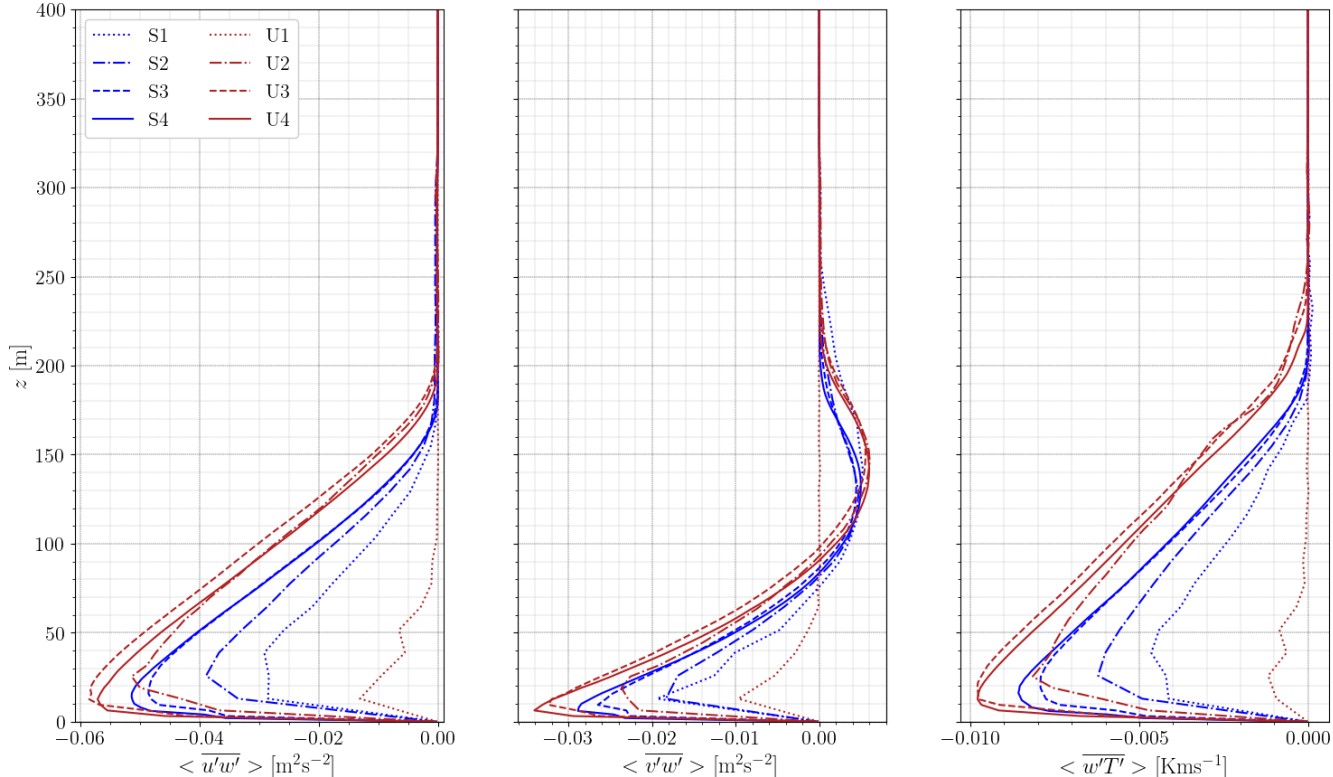

**Figure 14.** Time- and horizontally-averaged momentum and heat fluxes profiles.

To complement this analysis, the relative $L_2$ norm error is computed between horizontal average profiles from the present
work and the reference profiles in Beare et al. (2006). Results are summarized in Table 4. Excluding the coarsest cases, velocity
profile errors remain below $6\%$ and temperature profile errors below $0.1\%$. Grid convergence appears below $\Delta x = 6.25\,\mathrm{m}$, with
structured grids exhibiting slightly better agreement overall.

## 5  Conclusions

This study presents a high-order incompressible Navier–Stokes solver capable of performing large-eddy simulations of the
stable atmospheric boundary layer on unstructured meshes – a significant step forward, given the complexity of such simula-
tions. The solver incorporates the Coriolis force, the Boussinesq approximation for buoyancy effects, and wall modelling via
Monin–Obukhov similarity theory.

The framework was first validated against the Andrén benchmark, comparing mean profiles, variances, friction velocity,
and vertical momentum flux with previous studies. Simulations on both structured and unstructured grids produced similar
results, with only minor discrepancies. The unstructured grid slightly overestimated friction velocity and velocity variance,



primarily due to lower mesh quality near the wall and associated numerical diffusion. However, its influence was found to be less significant than that of the numerical solver choices.

The solver was then evaluated using the well-established GABLS1 benchmark. LES using both grid types at a recommended resolution of $\Delta x = 3.125\,\mathrm{m}$ reproduced the main features of the SBL with very good agreement compared to both the original and more recent studies. As with the neutral case, unstructured grids introduced marginally more numerical diffusion, and gradient estimation proved less accurate. This led to subtle differences in flux prediction and SBL evolution, with the boundary layer height in unstructured grids approximately $10\%$ higher, along with stronger momentum and heat fluxes. Nonetheless, these differences remain within the range of variability observed across other LES studies in the literature. Overall, subgrid-scale modelling, numerical schemes, and grid resolution were found to have a more pronounced influence on results than mesh structure.

A resolution sensitivity analysis further demonstrated that a grid spacing of $\Delta x = 6.25\,\mathrm{m}$ is sufficient to achieve boundary layer height predictions within $20\%$ of a high-resolution reference. The relative $L_2$ norm errors for horizontal velocity and temperature profiles remained below $6\%$ for both mesh types, with errors decreasing as resolution improved. These findings indicate that both structured and unstructured grids can provide robust and accurate LES of the SBL, particularly at $\Delta x = 3.125\,\mathrm{m}$ resolution.

To the best of the authors' knowledge, this work represents one of the first successful LES of the stable boundary layer using unstructured grids. This approach now enables high-fidelity simulations over geometrically complex terrain, where structured grids are impractical or inadequate. Future work will focus on extending this framework to such realistic applications in wind energy and atmospheric science.

## Appendix: GABLS1 source of errors

Two sources of debate can be highlighted in the design of the GABLS1 benchmark (Beare et al., 2006): initial condition definition and numerical errors accumulation.

### 1 Initial condition definition

The initial condition vertical temperature profile of the GABLS1 benchmark is spatially uniform, set to $T = 265\,\mathrm{K}$ from the ground up to $z = 100\,\mathrm{m}$ and then increases by $1\,\mathrm{K}/100\,\mathrm{m}$. To help the flow destabilization process, a random potential temperature perturbation of $0.1\,\mathrm{K}$ amplitude is superposed to the profile between $z = 0\,\mathrm{m}$ and $z = 50\,\mathrm{m}$. The definition of this random perturbation is left to the user's discretion, which is questionable. Commonly, users add a randomly generated noise on each control volume which is spatially uncorrelated. This can clearly have an impact on the flow evolution and will depends on the mesh resolution and grid partitioning.

To quantify its impact on the flow behaviour, two identical simulations based on the $\Delta x = 12.5\,\mathrm{m}$ structured grid are performed with the only difference being the random number seeds. Figure A1 shows the momentum and heat fluxes profiles spatially averaged over horizontal planes and temporally averaged between the $7th$ and the $8th$ hour, so long after initializa-





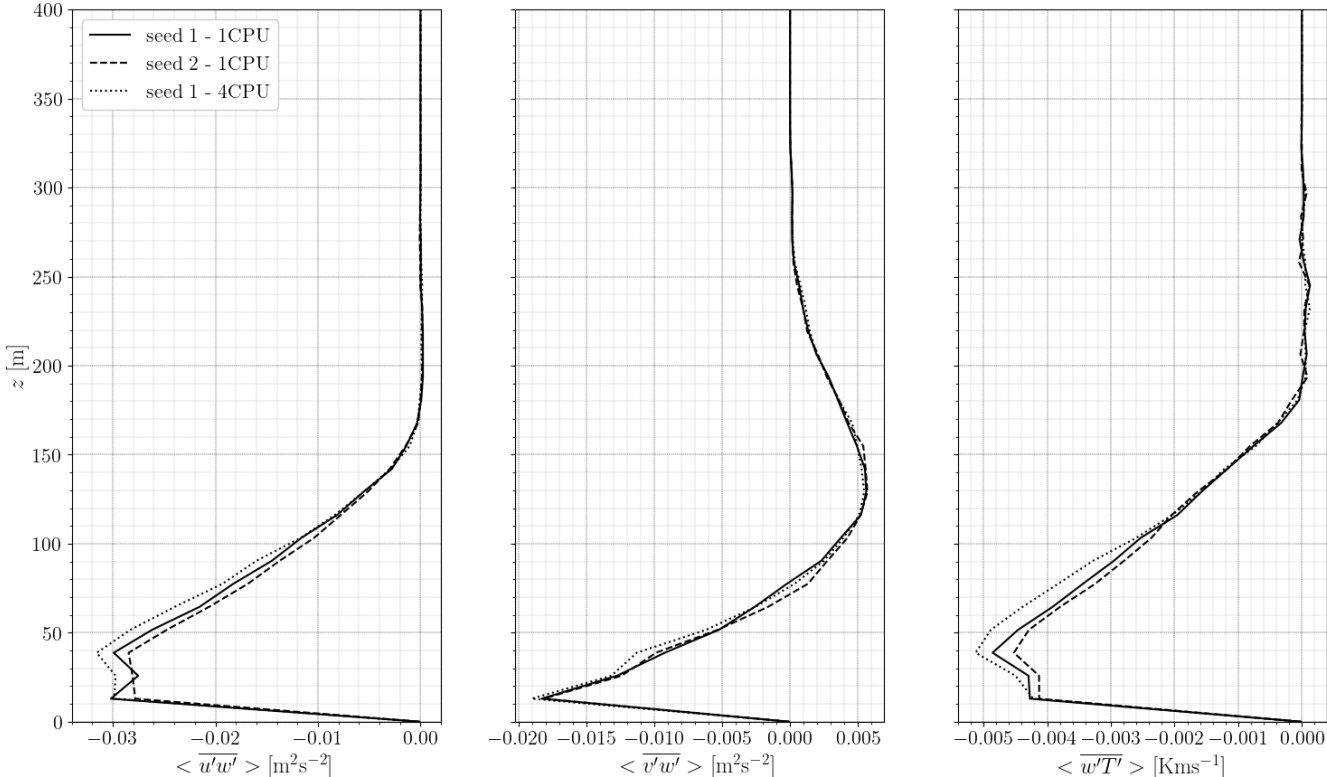

**Figure A1.** Time- and horizontally-averaged momentum and heat fluxes profiles on S3 mesh. Results with seed 1 and 1 CPU core (black solid line), seed 2 and 1 CPU Core (black dashed line) and seed 1 and 4 CPU cores (black dotted line).

tion. Results present a clear dependency on the random seed, with noticeable differences, showing a different flow evolution between the initialisation and the $8th$ hour. Similar gaps are observed for average velocity, temperature and velocity variance and these results are reproducible for different grid resolutions and numerical schemes, but not shown here for the sake of clarity.

This effect means that a small change in the initial profile affect the behaviour of the flow ans so the collected statistics. It can distort the comparison between codes since the random number generation will necessarily be different. Moreover, this random number is only determined by an amplitude and a mean, analogous to a white noise without spatial coherence. As different grid resolution were used in all GABLS1 studies, different fluctuation frequency were added. Since the flow behaviour is sensitive to this initial profile, part of the differences obtained when comparing two resolutions can be explained by this phenomenon. Similarly, it could also explain differences between structured and unstructured grids. Adding constraints on the random number, such as giving the fluctuation frequency or giving some spatial correlation, would help in having similar initial condition, whatever the mesh type and resolution. The perturbation would then be analogous to pink noise instead of white

noise. The results would still depend on the random number seed but at least would minimize differences when comparing different resolutions.

## A1 Numerical errors accumulation

Theoretically, a deterministic simulation behaviour is expected, since the resolution of the Navier-Stokes equations is fully deterministic. Simulations are reproducible and all states can be derived from the input data. However, numerical errors can
lead to non-deterministic flows, i.e. different results can be obtained with identical input data. The sources of numerical errors are various: node reordering, machine precision, operation orders, etc. In this respect, the grid partitioning and so the number of CPU cores used in a LES can cause variations in the results. It has been demonstrated that the propagation of numerical errors is linear for laminar flows but exponential for turbulent flows (Garciá, 2008). This difference between laminar and turbulent flows is due to the true chaotic nature of turbulence.
To illustrate this effect, two identical simulations were performed on the $\Delta x = 12.5\,\mathrm{m}$ structured grid with different number of CPU cores: one simulation with 1 CPU, the other with 4 CPUs and by keening the same random generator seed). Figure A1 shows the momentum and heat fluxes profiles for both cases. Momentum and heat fluxes profiles show discrepancies depending on the number of CPUs used. Similar gaps are observed for other quantities and is reproducible with other grid resolutions and numerical schemes but are not shown for the sake of brevity. As the errors accumulate quickly, working with higher machine
precision will not suppress the error propagation but only delay it. Since error propagation is exponential, the flow paths will always diverge (Garciá, 2008). To circumvent this effect, several simulations with different random number generations could be performed and averaged to give more statistical accuracy.

*Data availability.* All simulations data are fully available at https://doi.org/10.5281/zenodo.16884072. Commercial licenses for YALES2 can be purchased from CNRS.

*Author contributions.* UV performed the simulations, post-processing and was responsible for writing the paper. LV provided indispensable support in handling the simulation tool. All the authors provided valuable input and insights that were important to steer the work. They also proofread and amended the paper. PB and SZ were crucial is providing the necessary resources to produce this work.

*Competing interests.* UV, LV, MSS, PB and SZ declare that they have no conflict of interest.

*Acknowledgements.* This work has been initiated during the Extreme CFD Workshop & Hackathon (https://ecfd.coria-cfd.fr).



This project was provided with computer and storage resources by GENCI at TGCC thanks to the grant 2023-A0142A11335 on the supercomputer Joliot Curie's ROME partition and to CRIANN resources under the allocation 2012006.

     We acknowledge "Consortium des Équipements de Calcul Intensif" (CECI,Belgium), for awarding this project access to the LUMI super-computer, owned by the EuroHPC Joint Undertaking, hosted by CSC (Finland) and the LUMI consortium.

     The present research benefited from computational resources made available on Lucia, the Tier-1 supercomputer of the Walloon Region,
infrastructure funded by the Walloon Region under the grant agreement n°1910247.



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
