# Peer review of "Enabling the use of unstructured meshes for the Large Eddy Simulation of stable atmospheric boundary layers"

_Wind Energy Science, 2025_

## Referee Comment (RC1)

**Review: Enabling the use of unstructured meshes for the Large-Eddy Simulation of stable atmospheric boundary layers**

**Summary**

The authors present a validation study of the YALES2 finite-volume code for simulating atmospheric flow under neutral and stable conditions using unstructured grids. Results from structured and unstructured grids are evaluated against established benchmark studies, and grid-spacing sensitivity is also examined. The study concludes that unstructured grids can reproduce mean flow and turbulence statistics with reasonable fidelity, comparable to structured-grid results and observational data. Overall, the manuscript is well written and the findings are sound. However, the introduction and discussion would benefit from additional context on the respective strengths and limitations of unstructured versus structured grids, as well as further clarification of certain results. I recommend the manuscript for major revision.

**Major comments:**

1. One of the main motivations of this work is to accurately simulate atmospheric flow over complex terrain and under stably stratified conditions. There is a substantial body of literature demonstrating that LES codes with structured meshes have also been successfully used in such contexts. For example, finite-difference LES codes with both fixed (e.g., WRF-LES, EllipSys3D) and adaptive (e.g., AMR-Wind, ERF) grids has been applied extensively to atmospheric and wind turbine wake modeling under a wide range of stability and terrain conditions (e.g., Berg et al., 2018; Dar et al., 2019; Lattanzi et al., 2024). Some of these studies even include wind turbine simulations in regions with complex terrain and stably stratified flow conditions (e.g., Wise et al., 2022). The authors should acknowledge this body of literature. In addition, some of the statements regarding the limitations of structured meshes for simulating atmospheric flow over complex terrain (e.g., Lines 2–6, Line 338) come across as stronger than necessary. These could be rephrased more carefully or moderated to avoid overstating the case.

2. The results from Section 4.2.2 for the coarse unstructured grid (U1) merit further discussion. It is concerning to see such large deviations between U1 and S1. The time series in Figure 10 shows U1 does not reach stationarity. Moreover, the vertical structure that develops in U1 appears quite different from the other cases. It would be helpful if the authors could clarify whether these findings suggest a more general issue, namely if unstructured grids may be less reliable when the grid spacing is not small enough to resolve most of the inertial subrange of the flow? If smaller grid spacing is indeed required for unstructured grids to achieve results comparable to

structured grids, it would be useful for the reader to better understand what the practical advantages of unstructured grids are in this context. Also, the authors state in several places (e.g., Figure 3, Figure 7, Figure 8, Lines 248-250, Lines 252-254) that the unstructured grid can resolve more turbulence than the structured grid. However, Figure 14 shows that U1 very little turbulence, whereas S1is capable of resolving turbulence to some degree. Could the authors comment on why this is the case?

3. The comparison of turbulence fluxes and variances across grid resolutions could be clarified. As written, it appears that the manuscript reports only resolved turbulence fluxes. Since these are expected to vary substantially with grid resolution (from $\Delta x = 12.5$ m to $\Delta x = 2$ m), a more complete comparison of turbulence statistics in Section 4.2.2 should also include the SGS contributions. This would give a clearer picture of the total (resolved + modeled) turbulence stresses in the flow and how each grid performs. In addition, it would help readers if the authors could explicitly state whether the comparisons with observations and other LES codes are based on resolved fluxes alone or on total fluxes.

**Minor Comments:**

1. Indeed, unstructured grids offer great flexibility to adapt to complex geometries. However, in practice, high-resolution terrain data can be obtained 10-30 m resolution (Farr et al., 2007; USGS, 2021). So, how necessary is it to have unstructured grids if the underlaying terrain data is much coarser than the grid spacing of the model that is required to resolve turbulence in stably stratified flow? I recognize this is out of the scope of this manuscript, but I am curious about the authors' opinions.

2. Line 38: Wind veer also affects wake recovery (Abkar et al., 2018), wind turbine performance (Sanchez Gomez and Lundquist, 2020), and structural loads on turbines (Wu et al., 2025).

3. The authors should include the temperature equation from the model, especially since this manuscript focuses on thermally driven changes in turbulence and mean flow conditions.

4. Caption and labels in Figure 3: Please clarify if turbulence variances calculated using space and time averages (i.e., $\langle \bar{\cdot} \rangle$) or only time averages (i.e., $\bar{\cdot}$).

5. Figure 10 (bottom) and Figure 5 (bottom): The authors should consider re-scaling the panel for the Monin-Obukhov length so that differences in L are observable in the figure.

6. Appendix and associated discussion: The authors suggest that initial temperature perturbations contribute to simulation results and variability among models (Lines 229-233). I agree that if large enough perturbations are added, then the mean flow conditions may start to deviate. However, the results provided in the Appendix only

show differences in turbulence fluxes and not mean quantities. These differences are of the order of 5% and are largest above ~20 m. The authors should conduct a more thorough analysis before making such generalizations.

**References**

Abkar, M., Sørensen, J. N., and Porté-Agel, F.: An Analytical Model for the Effect of Vertical Wind Veer on Wind Turbine Wakes, Energies, 11, 1838, https://doi.org/10.3390/en11071838, 2018. in Figure 7

Berg, J., Troldborg, N., Menke, R., Patton, E. G., Sullivan, P. P., Mann, J., and Sørensen, N. N.: Flow in complex terrain - a Large Eddy Simulation comparison study, J. Phys.: Conf. Ser., 1037, 072015, https://doi.org/10.1088/1742-6596/1037/7/072015, 2018.

Dar, A. S., Berg, J., Troldborg, N., and Patton, E. G.: On the self-similarity of wind turbine wakes in a complex terrain using large eddy simulation, Wind Energ. Sci., 4, 633–644, https://doi.org/10.5194/wes-4-633-2019, 2019.

Farr, T. G., Rosen, P. A., Caro, E., Crippen, R., Duren, R., Hensley, S., Kobrick, M., Paller, M., Rodriguez, E., Roth, L., Seal, D., Shaffer, S., Shimada, J., Umland, J., Werner, M., Oskin, M., Burbank, D., and Alsdorf, D.: The Shuttle Radar Topography Mission, Reviews of Geophysics, 45, 2005RG000183, https://doi.org/10.1029/2005RG000183, 2007.

Lattanzi, A., Almgren, A., Quon, E., Natarajan, M., Kosovic, B., Mirocha, J., Perry, B., Wiersema, D., Willcox, D., Yuan, X., and Zhang, W.: ERF: Energy Research and Forecasting Model, 2024.

Sanchez Gomez, M. and Lundquist, J. K.: The effect of wind direction shear on turbine performance in a wind farm in central Iowa, Wind Energ. Sci., 5, 125–139, https://doi.org/10.5194/wes-5-125-2020, 2020.

USGS: United States Geological Survey 3D Elevation Program 1/3 arc-second Digital Elevation Model, https://doi.org/10.5069/G98K778D, 2021.

Wise, A. S., Neher, J. M. T., Arthur, R. S., Mirocha, J. D., Lundquist, J. K., and Chow, F. K.: Meso- to microscale modeling of atmospheric stability effects on wind turbine wake behavior in complex terrain, Wind Energ. Sci., 7, 367–386, https://doi.org/10.5194/wes-7-367-2022, 2022.

Wu, T., Cheng, Y., Sun, Y., and Zhang, J.: Study on load distribution characteristics and wind-resistant performance of standstill wind turbines considering the effect of wind veer, Renewable Energy, 254, 123726, https://doi.org/10.1016/j.renene.2025.123726, 2025.

---

## Author Comment (AC1)

**Enabling the use of unstructured meshes for the Large Eddy Simulation of stable atmospheric boundary layers**

Ulysse Vigny [1,2], Léa Voivenel [3], Mostafa Safdari Shadloo [2,4], Pierre Bénard [2], and Stéphanie Zeoli [1]

[1]Fluid Machine Unit, University of Mons (UMONS), Mons, 7000, Belgium
[2]INSA Rouen Normandie, Univ Rouen Normandie, CNRS, Normandie Univ, CORIA UMR 6614, F-76000 Rouen, France
[3]CNRS, INSA Rouen Normandie, Univ Rouen Normandie, Normandie Univ, CORIA UMR 6614, F-76000 Rouen, France
[4]Institut Universitaire de France, Rue Descartes, F-75231 Paris, France

**Correspondence:** Ulysse Vigny  (ulysse.vigny@umons.ac.be)

First and foremost, the authors wish to sincerely thank the reviewers for their time, review, and appreciation of the work. We have addressed all reviewers' comments below and in the revised paper, allowing to clarify parts of the paper.

**Anonymous Referee #1**

*Comment 1*: *One of the main motivations of this work is to accurately simulate atmospheric flow over complex terrain and under stably stratified conditions. There is a substantial body of literature demonstrating that LES codes with structured meshes have also been successfully used in such contexts. For example, finite-difference LES codes with both fixed (e.g., WRF-LES, EllipSys3D) and adaptive (e.g., AMR-Wind, ERF) grids has been applied extensively to atmospheric and wind turbine wake modeling under a wide range of stability and terrain conditions (e.g., Berg et al., 2018; Dar et al., 2019; Lattanzi et al., 2024). Some of these studies even include wind turbine simulations in regions with complex terrain and stably stratified flow conditions (e.g., Wise et al., 2022). The authors should acknowledge this body of literature. In addition, some of the statements regarding the limitations of structured meshes for simulating atmospheric flow over complex terrain come across as stronger than necessary. These could be rephrased more carefully or moderated to avoid overstating the case.*

The authors agree on the limited literature review of the initial version. Articles demonstrating that LES codes with structured meshes have been successfully used in the wind energy context and have been acknowledged in the revised version. A new paragraph in the introduction is now dedicated to this topic, citing both the topology conforming structured mesh technique and the IBM approach. The statements have been re-examined. The use of unstructured meshes is proposed as a new approach rather than being presented as the only option.

*Comment 2*: *The results from Section 4.2.2 for the coarse unstructured grid (U1) merit further discussion. It is concerning to see such large deviations between U1 and S1. The time series in Figure 10 shows U1 does not reach stationarity. Moreover, the vertical structure that develops in U1 appears quite different from the other cases. It would be helpful if the authors could*

*clarify whether these findings suggest a more general issue, namely if unstructured grids may be less reliable when the grid spacing is not small enough to resolve most of the inertial subrange of the flow? If smaller grid spacing is indeed required for unstructured grids to achieve results comparable to structured grids, it would be useful for the reader to better understand what the practical advantages of unstructured grids are in this context. Also, the authors state in several places (e.g., Figure 3, Figure 7, Figure 8, Lines 248-250, Lines 252-254) that the unstructured grid can resolve more turbulence than the structured*

*grid. However, Figure 14 shows that U1 very little turbulence, whereas S1 is capable of resolving turbulence to some degree. Could the authors comment on why this is the case?*

Our initial explanation could indeed have suggested that the observed differences were of physical origin and related to turbulence resolution, whereas they are, in fact, primarily numerical. To highlight this, the turbulent kinetic energy spectra at two different vertical locations are plotted in Fig. 1, estimated using the Welch method (Welch, 1967), with Hamming windows and 25% overlap. At both heights, the spectral content matches between mesh types. The largest flow structures, corresponding to low frequency, are well represented. Minor differences can be noted at higher frequencies where unstructured grid simulation exhibits a spectra that decrease faster than the structured ones, resulting in a lightly smaller maximum wave number. This discrepancy is probably due to the marginally smaller number of control volumes in the unstructured mesh.

Velocity variance differences are thus not the result of inaccuracies in the energy cascade resolution.

The spectra figure are added to the revised version of the paper.

[Figure]

**Figure 1.** Energy spectra for structured (S3) and unstructured (U3) meshes at two different heights.

An additional difficulty for the unstructured mesh consists in the evaluation of wall gradient needed to compute wall fluxes. Such wall normal gradient is particularly sensitive to mesh quality and orientation, particularly stressed when grid cells are irregular. This leads to inaccuracy in heat and momentum transfer calculations. To give a more practical explanation, identifying the index of the first fluid node connected to a given wall face would require a dedicated connectivity table. Instead, we reconstruct the value at the first node by means of a Taylor expansion between the face center and the cell centroid, as illustrated by Fig. 2. When the cell quality is poor, this reconstruction can lead to a biased estimate of the value at the first node.

[Figure]

**Figure 2.** First node value estimation in unstructured mesh.

In addition, the vertical position of the first fluid node may vary from one cell to another, resulting in inhomogeneities in cell-to-cell momentum fluxes. This behaviour can be mitigated through the filtering procedure mentioned in the paper. This filtering, which can be regarded as a local averaging among immediate neighbouring cells, tends to lose its meaning for very coarse meshes, where the height of the first cell may vary significantly from one cell to another. Specifically for the U1 case, the wall gradient, and hence the corresponding friction velocity, is severely underestimated, which results in markedly different flow dynamics, in this case leading to a flow that remains laminar.

*Comment 3*: *The comparison of turbulence fluxes and variances across grid resolutions could be clarified. As written, it appears that the manuscript reports only resolved turbulence fluxes. Since these are expected to vary substantially with grid resolution (from $\Delta x = 12.5m$ to $\Delta x = 2m$), a more complete comparison of turbulence statistics in Section 4.2.2 should also include the SGS contributions. This would give a clearer picture of the total (resolved + modeled) turbulence stresses in the flow and how each grid performs. In addition, it would help readers if the authors could explicitly state whether the comparisons with observations and other LES codes are based on resolved fluxes alone or on total fluxes.*

To address the concern regarding the comparison of turbulent fluxes across grid resolutions, we have included in Fig. 3 the total contributions of the turbulent fluxes $\langle \overline{u'w'} \rangle$ and $\langle \overline{v'w'} \rangle$ for S3 and U3 resolution. A comparative analysis of the modelled regions in both mesh configurations confirms that the observed discrepancies are primarily attributable to the wall flux computation rather than to mesh resolution. Furthermore, the proportion of the flux captured by the modelled (SGS) component remains significantly lower than the resolved one, reinforcing the conclusion that the differences are not due to under-resolution but rather by the treatment of near-wall dynamics.

[Figure]

**Figure 3.** Comparison of total, resolved, and SGS fluxes of $\langle \overline{u'w'} \rangle$ and $\langle \overline{v'w'} \rangle$ for S3 and U3.

*Comment 4: Indeed, unstructured grids offer great flexibility to adapt to complex geometries. However, in practice, high-resolution terrain data can be obtained 10-30 m resolution (Farr et al., 2007; USGS, 2021). So, how necessary is it to have unstructured grids if the underlaying terrain data is much coarser than the grid spacing of the model that is required to resolve*
*turbulence in stably stratified flow? I recognize this is out of the scope of this manuscript, but I am curious about the authors' opinions.*

From our point of view, the use of unstructured meshes could be advantageous for four reasons. First, for steep complex terrain, the refinement of structured meshes seems limited by orthogonality issues. In such scenario, finer unstructured meshes could be used. Second, unstructured meshes can intrinsically follow complex geometries like building structures. There is no
need for additional technique like IBM to consider such as urban flow, which is an advantage. Third, generating body-fitted meshes that follow complex geometries is easier with unstructured grids, compared to structured ones, and requires less human effort. An automatic unstructured mesh generation tool that follows topography has been developed in the YALES2 platform, enabling such meshes to be obtained simply and quickly. Fourth, using unstructured meshes allows less restrictive adaptive mesh refinement than structured mesh. This feature is particularly relevant in wind turbine simulations, where AMR can be
used in wind turbine wake (Zeoli et al., 2020, Journal of Physics: Conference Series ; Vigny et al., 2020, Journal of Physics: Conference Series)

*Comment 5: Line 38: Wind veer also affects wake recovery (Abkar et al., 2018), wind turbine performance (Sanchez Gomez and Lundquist, 2020), and structural loads on turbines (Wu et al., 2025).*

We fully agree with this comment. The text has been modified accordingly.

*Comment 6: The authors should include the temperature equation from the model, especially since this manuscript focuses on thermally driven changes in turbulence and mean flow conditions.*

The authors apologize for this unfortunate omission. The temperature equation have been naturally introduced in the revised
version of the paper.

*Comment 7: Caption and labels in Figure 3: Please clarify if turbulence variances calculated using space and time averages or only time averages.*

Turbulence variances are calculated using space and time averages. Clarifications have been added to the text as well as into
the legend and label of the figure.

*Comment 8: Figure 10 (bottom) and Figure 5 (bottom): The authors should consider re-scaling the panel for the Monin-Obukhov length so that differences in L are observable in the figure.*

Indeed, a re-scaling of the Monin-Obukhov length would allow us to observe differences in the Moning-Obukhov length, $L$.
However, Fig. 10 aims to show the overall temporal evolution of frictional velocity, wall heat flux and Monin-Obukhov length. This provides an overview of the flow destabilisation process discussed in the text. A re-scaling would prevent this demonstration. In addition, $L$ is only a resultant of the wall heat flux and the frictional velocity $L \sim u_*^3/Q_w$. Therefore, evolution of $L$ is secondary. Finally, to measure the boundary layer height, the vertical distribution of turbulent stress is used (Kosovic and Curry, Journal of atmospheric sciences, 2000), as shown in Tab. 3. Thus, the Monin-Obukhov length is not directly used in this
study.

*Comment 9: Appendix and associated discussion: The authors suggest that initial temperature perturbations contribute to simulation results and variability among models (Lines 229-233). I agree that if large enough perturbations are added, then the mean flow conditions may start to deviate. However, the results provided in the Appendix only show differences in turbu-*
*lence fluxes and not mean quantities. These differences are of the order of $5\%$ and are largest above $20\,\mathrm{m}$. The authors should conduct a more thorough analysis before making such generalizations.*

Mean quantities have been added in the appendix (Fig. A1). It is true that differences are not visible in the averaged velocity components and temperature profiles. Only light differences are shown by the flux profiles (Fig. A2). The text comments have been moderated to represent this analysis. However, the authors wish to emphasise that still obtaining differences for spatially
and temporally averaged quantities demonstrates a real sensibility to numerical inputs, leading to flow dynamics difference. To the author's knowledge, this impact is not raised in the literature of this case.

**Anonymous Referee #2**

*Comment 1: In the manuscript the validation of an unstructured LES code is presented in clear and logical way. The valida-tion of an unstructured grid model can be of interest for the wind energy community. However, the selected test cases are very basic horizontally homogeneous atmospheric boundary layers (ABLs): a neutrally stratified atmospheric boundary layer case based on Andren (1994) paper, and a stably stratified case based on GABLES 1 intercomparison study (Beare et al., 2006). While these cases demonstrate the ability of the model to generally reproduce the structure of these ABLs, they do not demon-*

*strate the advantage of using unstructured grids. Unstructured grids could be advantageous for simulation of flows in complex terrain or around structures (e.g., urban flow simulations). However, structured grids have been successfully used for such simulations when combined with immersed boundary or immersed force approaches (e.g., Lundquist et al. 2009, MWR; Arthur et al. 2018, MWR; Muñoz-Esparza et al. 2020, JAMES). When also combined with mesh refinement, structured grid models can be used to resolve structures in high detail (e.g., Energy Research and Forecasting model, https://github.com/erf-model/ERF;*

*Lattanzi et al. 2024, arXiv, to appear in JAMES). The authors do not address this possibility and contrast unstructured grids with structured grids including immersed boundary approach and mesh refinement.*

A new paragraph in the introduction has been added demonstrating that LES codes with structured meshes have been suc-cessfully used in such context. The use of unstructured meshes is proposed as a new approach. It is true that the selected test cases do not highlight the advantage of using unstructured grids. However, it is a first validation step. From our point of view, it was important to compare results from structured and unstructured meshes on relatively simple test cases to reveal potential differences such as flux estimation near the wall before tackling more intricate studies. However, the ultimate goal of using unstructured meshes is to perform complex terrain studies without terrain smoothing step or grid orthogonality issues. In addi-tion, the generation of an unstructured mesh in this type of scenario will require less human effort.

*Comment 2: In the neutrally stratified case, the unstructured model produces higher velocity variances, while in stably stratified case it also results in higher turbulent stresses. The authors attribute this to "increased resolved turbulence, which can be attributed to differences in near-wall resolution and numerical dissipation." This argument is not convincing; a reason for differences in near-wall resolution should be provided, also, it is not clear why would numerical dissipation be lower for the unstructured grid with the same number of degrees of freedom and the same numerical scheme as on the structured grid.*

*What is missing is the spectral analysis to demonstrate that unstructured grid can accurately reproduce energy cascade and that the enhanced variances are not result of inaccuracies in the energy cascade from large to small scales. While spectral analysis of unstructured grids is not straight forward – it can be accomplished either by interpolation (e.g., Juricke et al. 2022, JAMES) or by analyzing time series at a point (or better many points in space). The differences between results obtained using the structured and the unstructured grid should not be attributed to supposedly lower numerical dissipation without evidence.*

*Furthermore, it would be important to provide comparison of computational performance of the unstructured and structured models for the same number of degrees of freedom.*

The authors fully agree with the reviewer. Our initial explanation could indeed have suggested that the observed differences were of physical origin (related to turbulence resolution), whereas they are, in fact, primarily numerical. Indeed, the spectra computation at two different heights *e.g.* $z = 50\,\mathrm{m}$ and $z = 100\,\mathrm{m}$ show that the spectral content for both the structured and
unstructured meshes are quite similar, Fig. 1.

As specified in the answer of Reviewer #1-Comment 2, identifying the index of the first fluid node connected to a given wall face in an unstructured mesh would require a dedicated connectivity table. Instead, we choose to reconstruct the value at the first node by means of a Taylor expansion between the face center and the cell centroid, as illustrated in Fig. 2 of this reply. When the cell quality is poor, this reconstruction can lead to a biased estimate of the value at the first node.

In addition, the vertical position of the first fluid node may vary from one cell to another, resulting in "spotty" momentum fluxes. However, this behaviour can be mitigated through the filtering procedure mentioned in the paper. The coarser the mesh, the more pronounced these variations may become.

Concerning the computational performance, an overcost of $14\%$ is measured for the U3 mesh case compared to the S3 one, corresponding to the recommended resolution. This noticeable increase remains evaluated on a simple bi-periodic 3D box configuration. For more realistic applications with complex topography, the comparison is not relevant since a body-fitted structured mesh approach cannot be used.

*Comment 3: Equation 1 – Since the model is incompressible and Boussinesq approximation is used it would be important to include the equation for the potential temperature and replace density with potential temperature.*

The authors apologize for the unfortunate omission of the potential temperature equation. The temperature equation have been naturally introduced in the revised version of the paper. The text has been modified accordingly.

*Comment 4: Line 91 – It should be "Absolute value of the Obukhov lengths," since in unstably stratified case the length scale is negative.*

We agree with the remark and the text has been modified accordingly.

*Comment 5: Subsection 2.3 – Number of degrees of freedom (nodes) should be given for both, structured and unstructured grids.*

Grid sizes, number of elements and number of nodes for the Andren case has been added in section 3.1. Concerning the GABLS1 case, these informations were already given in section 4.1.

*Comment 6: Line 129 – Since the model is incompressible, constant (and uniform) density, it is not clear what is meant by "reference density." It should be a reference temperature.*

The reference density $\rho_0$ corresponds to the "Boussinesq" base-state (constant) density defined such as $\frac{\theta - \theta_0}{\theta_0} = -\frac{\rho - \rho_0}{\rho_0}$

*Comment 7: Line 150 – Elevated values of the friction velocity in case of the unstructured grid are attributed to increased numerical diffusion, however, later the elevated levels of velocity variances in simulations with the unstructured grid are attributed to lower numerical dissipation. These two statements cannot be easily reconciled. It is not obvious if they both can*

*hold simultaneously. The authors should provide clear explanation.*

A more accurate explanation for these differences in behaviour has been given in comment #2. The article content has also been modified accordingly.

**Comment 8**: *Figure 3 – Shown is only resolves turbulent stress since it is zero at the surface. It would be important to plot total turbulent stress, resolved and subgrid as it is commonly done (e.g., Figure 7 a) in Chow et al. 2005, JAS, cited in the*

*manuscript).* **Comment 9** : *Figure 8 – Same as for Figure 3.* **Comment 10** : *Figure 14 – Same as for Figure 3.*

To address the concern regarding the comparison of turbulent fluxes between structured and unstructured grids, we have included in Fig. 3 the total contributions of the turbulent fluxes $\langle \overline{u'w'} \rangle$ and $\langle \overline{v'w'} \rangle$ for S3 and U3 meshes. To preserve computational resources, we focus on the stable configuration of the GABLS1 case at the recommended resolution. This setup presents a more stringent challenge for turbulence modelling compared to the neutral case. The proportion of the flux captured by the modelled (SGS) component remains significantly lower than the resolved one. The results indicate that the contribution of the SGS model is more pronounced near the surface, which aligns with findings from previous studies. However, the modelled flux values are comparable across both mesh types, suggesting that the observed differences stem from the resolved component and are mainly due to numerical treatment (see comment 2 for further details). The graphs and associated commentary have been integrated into the revised manuscript.